# Stratospheric ozone loss over the Eurasian continent induced by the polar vortex shift

Jiankai Zhang [1], Wenshou Tian[1], Fei Xie[2], Martyn P. Chipperfield [3], Wuhu Feng [3,4], Seok-Woo Son[5], N. Luke Abraham [6,7], Alexander T. Archibald[6,7], Slimane Bekki[8], Neal Butchart[9], Makoto Deushi[10], Sandip Dhomse [3], Yuanyuan Han[1], Patrick Jöckel[11], Douglas Kinnison[12], Ole Kirner[13], Martine Michou[14], Olaf Morgenstern[15], Fiona M. O'Connor[9], Giovanni Pitari[16], David A. Plummer[17], Laura E. Revell[18,19], Eugene Rozanov[18,20], Daniele Visioni[16,21], Wuke Wang[22] & Guang Zeng[15]

The Montreal Protocol has succeeded in limiting major ozone-depleting substance emissions, and consequently stratospheric ozone concentrations are expected to recover this century. However, there is a large uncertainty in the rate of regional ozone recovery in the Northern Hemisphere. Here we identify a Eurasia-North America dipole mode in the total column ozone over the Northern Hemisphere, showing negative and positive total column ozone anomaly centres over Eurasia and North America, respectively. The positive trend of this mode explains an enhanced total column ozone decline over the Eurasian continent in the past three decades, which is closely related to the polar vortex shift towards Eurasia. Multiple chemistry-climate-model simulations indicate that the positive Eurasia-North America dipole trend in late winter is likely to continue in the near future. Our findings suggest that the anticipated ozone recovery in late winter will be sensitive not only to the ozone-depleting substance decline but also to the polar vortex changes, and could be substantially delayed in some regions of the Northern Hemisphere extratropics.

[1] Key Laboratory for Semi-Arid Climate Change of the Ministry of Education, College of Atmospheric Sciences, Lanzhou University Lanzhou 730000 Gansu, China. [2] College of Global Change and Earth System Science, Beijing Normal University, 100875 Beijing, China. [3] Institute for Climate and Atmospheric Science, School of Earth and Environment, University of Leeds, Leeds LS2 9JT, UK. [4] National Centre for Atmospheric Science, University of Leeds, Leeds LS2 9JT, UK. [5] School of Earth and Environmental Sciences, Seoul National University, 1 Gwanak-ro, Gwanak-gu, Seoul 08826, South Korea. [6] National Centre for Atmospheric Science, University of Cambridge, Cambridge CB2 1EW, UK. [7] Department of Chemistry, University of Cambridge, Cambridge CB2 1EW, UK. [8] LATMOS, Université Pierre et Marie Curie, 4 Place Jussieu Tour 45, couloir 45-46, 3e étage Boite 102, 75252 Paris Cedex 05, France. [9] Met Office Hadley Centre, FitzRoy Road, Exeter EX1 3PB, UK. [10] Meteorological Research Institute, 1-1 Nagamine Tsukuba, Ibaraki 305-0052, Japan. [11] Deutsches Zentrum für Luft- und Raumfahrt e.V. (DLR), Institut für Physik der Atmosphäre Münchner Strasse 20, Oberpfaffenhofen, 82234 Wessling, Germany. [12] Atmospheric Chemistry Observations and Modeling Laboratory, National Center for Atmospheric Research, 3090 Center Green Drive, Boulder, CO, 80301, USA. [13] Karlsruhe Institute of Technology, Steinbuch Centre for Computing, P.O Box 3640, 76021 Karlsruhe, Germany. [14] METEO-FRANCE CNRM, 42 Avenue G. Coriolis, 31057 Toulouse, France. [15] National Institute of Water and Atmospheric Research, 301 Evans Bay Parade, Hataitai, Wellington 6021, New Zealand. [16] Dipartimento di Scienze Fisiche e Chimiche Via Vetoio, Università dell'Aquila, 67100 L'Aquila, Italy. [17] Climate Research Branch, Environment and Climate Change Canada, 550 Sherbrooke St. West, West Tower, 19th Floor, Montreal, QC, H3A 1B9, Canada. [18] Institute for Atmospheric and Climate Science, ETH Zürich (ETHZ), Universitätstrasse 16, 8092 Zürich, Switzerland. [19] Bodeker Scientific, 42 Russell Street, Alexandra 9320, New Zealand. [20] Physikalisch-Meteorologisches Observatorium Davos–World Radiation Center (PMOD/WRC), Dorfstrasse 33, 7260 Davos Dorf, Switzerland. [21] CETEMPS, Universitá dell'Aquila, 67100 L'Aquila, Italy. [22] Institute for Climate and Global Change Research, School of Atmospheric Sciences, Nanjing University, Jiangsu, 210023 Nanjing, China. Correspondence and requests for materials should be addressed to W.T. (email: wstian@lzu.edu.cn)

Stratospheric ozone protects life on Earth by strongly absorbing harmful solar ultraviolet (UV) radiation[1–3]. It also plays an important role in modulating the global climate system by partly controlling large-scale atmospheric circulations via its radiative impact and radiative-chemical-dynamical feedbacks, which are seen in both the Southern Hemisphere[4–8] and the Northern Hemisphere[9–12]. Global stratospheric ozone concentrations have experienced persistent declines in response to the increased ozone-depleting substance (ODS) levels in the 20th century. Owing to the effects of the Montreal Protocol and its amendments on ODS levels, stratospheric ozone depletion has recently stabilized and begun to reverse[13–16]. Particularly, an increase in Antarctic ozone has been reported to be linked to ODS reductions[17]. In contrast, the Arctic polar vortex is more dynamically disturbed by planetary waves than its Antarctic counterpart[18–20], and complex interactions between chemical and dynamical processes make it more challenging to identify Arctic stratospheric ozone trends[21–26]. Some studies have argued that dynamical changes such as Arctic polar vortex changes will play a more significant role in affecting Arctic ozone; consequently, the projected timing of Arctic stratospheric ozone recovery is not robust[26–30].

It is well documented that the dynamics of the Arctic polar vortex has a significant impact on the spatio-temporal distributions of stratospheric ozone and other tracer gases[31–33]. Many studies have shown that an anomalously cold polar vortex in late winter could enhance springtime ozone depletion through the increased efficiency of heterogeneous reactions[27,32,33]. The polar vortex itself can also affect mid-latitude ozone levels through the ozone dilution process, i.e., mixing of polar ozone-poor air with mid-latitude ozone-rich air[34–36]. Knudsen and Andersen[37] pointed out that the springtime ozone depletion in the northern middle latitudes during the 1979–1997 period, which maximized over Europe and Russia, is partly caused by long-term changes in the strength of the polar vortex. Recent studies have speculated that the future evolution of Northern Hemisphere stratospheric ozone may be more driven by changes in the strength of Arctic polar vortex than by chemical forcings (i.e., decreasing ODS levels from the continuing implementation of Montreal Protocol)[26–28].

Zhang et al.[38] recently reported that Arctic sea-ice loss has contributed to a persistent late-winter shift of the Arctic vortex towards the Eurasian continent over the past three decades. However, its impact on regional ozone changes remains unclear. The present study addresses this issue by analysing observed total column ozone (TCO) with a chemical transport model (SLIM-CAT[39], see details in Methods section and Supplementary Fig. 1) and chemistry-climate model (CCM) simulations. Our results reveal that the polar vortex shift towards the Eurasian continent has significantly enhanced the February TCO loss over this continent in the past three decades; this phenomenon may continue in the coming decades.

## Results

**EOF analysis of total column ozone.** To quantify long-term variations of TCO in the Northern Hemisphere extratropics, we first perform an Empirical Orthogonal Function (EOF) analysis of February TCO from the MSR2 data set[40] (Fig. 1a, c, e) and SLIMCAT (Fig. 1b, d, f) for the period of 1980–2012. Only February is considered here because the polar vortex shift towards Eurasia is the strongest during this month[38]. The leading mode accounts for 41.5% of the total variance in MSR2, and shows a uniform TCO change covering northern mid- and high latitudes (Fig. 1a). This mode largely reflects the TCO variation attributed to the strength of the polar vortex, i.e., the first principal component (PC1) time series has a strong positive correlation with

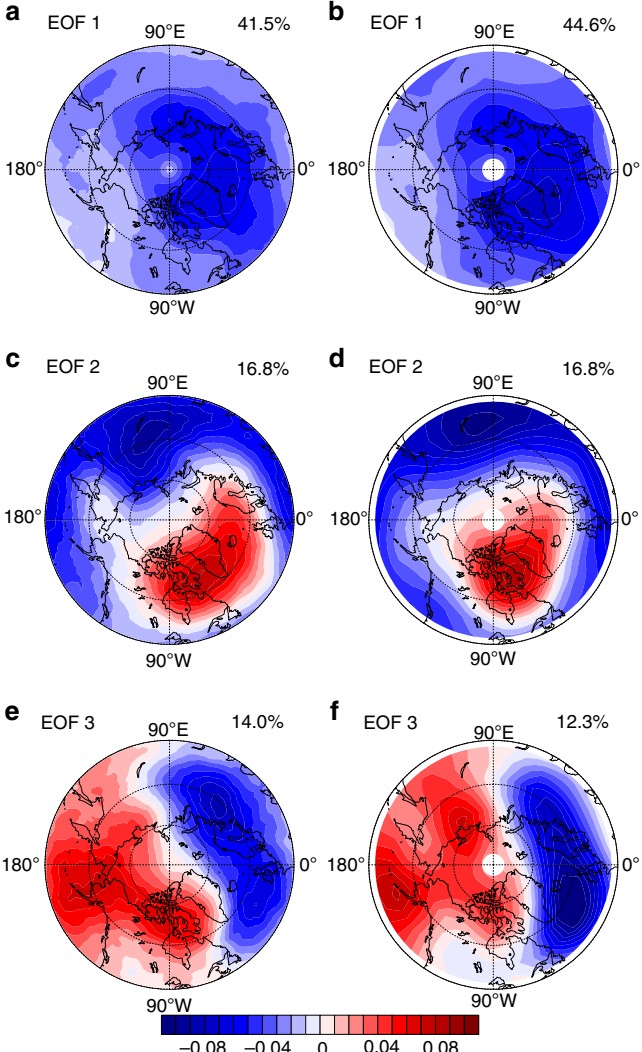

**Fig. 1** EOF of total column ozone over the Northern Hemisphere extratropics. Spatial patterns of EOF1, EOF2 and EOF3 of February mean total column ozone (TCO) over 45–90°N derived from **a**, **c**, **e** MSR2 data and **b**, **d**, **f** SLIMCAT full chemistry simulation for the period 1980–2012. The percentage of explained variance is shown in the top right of each plot. The minimum latitude of the polar stereographic projections is 45°N. The values in 1987, 2006 and 2009 are not included for EOF analysis because the polar vortex broke up and its shape was distorted during February in these years (Methods)

the polar vortex strength, defined as potential vorticity (PV) averaged between 430 and 600 K within 65–90°N (the correlation coefficient between PC1 and this average PV index is >0.7; Supplementary Fig. 2a). This can be explained by more efficient heterogeneous reactions leading to intensified ozone chemical depletion and suppression of the ozone influx through the vortex boundaries in case of a stronger, colder and more isolated polar vortex.

Unlike EOF1, both EOF2 and EOF3 exhibit a clear dipolar spatial pattern; their principal components explain 16.8% and 14.0% of the February TCO variance (Fig. 1c, e), respectively. More importantly, both PC2 and PC3 correlate well with structural change of the polar vortex, i.e., they are correlated with the polar vortex shift index defined by Zhang et al.[38] (Supplementary Fig. 2c, e; see Methods section for the definition of polar vortex shift index). Given this common correlation, we

combined EOF2 and EOF3 into a single mode (EOF2 + EOF3, Fig. 2a), which shows a clear signature of the spatial pattern with a negative centre of TCO variation over Eurasia and a positive one over North America. This mode is hence referred to as the Eurasia-North America dipole (ENAD) thereafter. Its spatial pattern well resembles the dipolar structure in PV variations associated with a polar vortex shifting towards the Eurasian continent (Fig. 1a in Zhang et al.[38]), and is remarkably similar to the regression pattern of TCO against the vortex shift index (Fig. 2g) with a strong spatial correlation up to 0.85. The combined PC (PC2 + PC3, Fig. 2d) is also better correlated with the polar vortex shift index than individual PC2 and PC3 time series, with a correlation coefficient of 0.59, that is statistically significant at the 99% confidence level. A similar ENAD mode is also found in other data sets such as NIWA[41] and ECMWF-Interim[42] ozone (Supplementary Fig. 3).

Both EOF1 and ENAD mode (and their PC time series) are well reproduced by the SLIMCAT off-line chemical transport model simulation (Fig. 1b, d, f, and Supplementary Fig. 2b, d, f; see Methods section for the details of the simulations). The correlation coefficient between the simulated PC of the ENAD mode and the vortex shift index (Fig. 2e) is 0.75, larger than that derived from observations. The spatial correlation between the ENAD mode and the TCO regressed against the polar vortex shift is up to 0.90 in the model simulations (Fig. 2b, h). These similarities allow us to use the SLIMCAT model to attribute the ENAD-related ozone change to dynamical and chemical ozone changes.

It is important to note from Fig. 2d, e that the PC of the ENAD mode, which explains ~30% of interannual variance of February TCO, shows an increasing trend over the last three decades. This trend is statistically significant, indicating that the polar vortex shift may cause not only interannual variability of regional TCO but also a decreasing trend in TCO over the Eurasian continent that would delay the TCO recovery caused by ODS reduction.

More evidence that the ENAD can affect the environment is found in Fig. 2c, which indicates that the ENAD-like mode also exists in the clear-sky UV radiation over the Northern Hemisphere. As expected, this mode is largely in anti-phase with the ENAD-related TCO anomalies (compare Fig. 2a, c), and its PC time series is significantly correlated with the polar vortex shift index (Fig. 2f). The ENAD-like mode in the UV index also resembles the regression pattern of the UV index against the vortex shift index (Fig. 2i), suggesting that the polar vortex shift can exert a significant influence on the clear-sky UV radiation over the Northern Hemisphere.

**Impacts of polar vortex shift on the stratospheric ozone**. To better understand the effects of the polar vortex shift on ozone changes over the Eurasian continent, we analyse the relationship between PV changes, dynamical ozone changes, and chemical ozone changes within the 430–600 K isentropic layers during February of 2001, 2008, and 2010 in the SLIMCAT model simulations. These 3 years are chosen because they have undergone the largest polar vortex shifts (Fig. 2d, e). Figure 3a–c show the isentropic PV anomalies in February for these 3 years with respect to the climatology. There are positive PV anomalies over the Eurasian continent but negative PV anomalies over the North American continent, indicating that the polar vortex is shifted towards Eurasia compared with the climatological polar vortex[38]. This shift is seen at all isentropic levels in the lower stratosphere (Supplementary Fig. 4). Note that the polar vortices in 2001 and 2010 tend to shift towards the sector 90–120°E (Fig. 3a, c), while the polar vortex in 2008 shifts towards the sector 90–60°E (Fig. 3b). The patterns of the PV anomalies in 2001 and 2010 are

similar to EOF2 while the 2008 pattern looks more like EOF3. Correspondingly, the ENAD, which is obtained by the sum of EOF2 and EOF3, essentially reflects the influence of polar vortex shifting towards the Eurasian continent (0–120°E sector).

In the SLIMCAT simulation, the transport contribution to ozone anomalies (hereinafter referred to as dynamical ozone anomalies) can be calculated from a passive ozone tracer, which is transported by the model winds, while the heterogeneous chemistry contribution can be calculated from the difference between the ozone concentrations simulated with full chemistry and those simulated without heterogeneous chemistry (Methods). Figure 3d–f clearly show negative dynamical ozone anomalies over the Eurasian continent with the low ozone centre being also shifted, in agreement with the polar vortex shift. Note that in isentropic coordinates, areas with high PV typically represent the air coming from the polar region. Dynamical ozone anomalies, however, do not entirely explain the whole ozone anomalies. There are also strong negative chemical ozone anomalies over the Eurasian continent, implying that the polar vortex shift enhances heterogeneous chemical (HC) ozone loss over this region (Fig. 3g–i). In fact, all centres of HC ozone loss lie within the vortex edge, away from the pole, and move towards the Eurasian continent in association with the polar vortex shift. The active chlorine ($ClO_x = Cl + ClO + 2 \times Cl_2O_2$) concentration, an indicator of the chlorine-related heterogeneous chemistry inside the vortex, also increases (Supplementary Fig. 5) in accord with the enhanced chemical ozone loss over Eurasia.

The relationship between the anomalies in PV and key tracers over Eurasia is further examined in Fig. 4. A statistically significant negative correlation between PV anomalies and dynamical ozone loss is again obvious (Fig. 4a), confirming the close linkage between the polar vortex shift and dynamical ozone depletion. In addition, high PV anomalies over the Eurasian continent correspond to anomalously strong ozone loss from heterogeneous chemistry (Fig. 4b). When the PV anomalies over Eurasia are larger than normal, $ClO_x$ concentrations are anomalously high (Fig. 4c) and the ratio of hydrogen chloride (HCl) to total inorganic chlorine ($Cl_y = Cl + 2 \times Cl_2 + ClO + OClO + 2 \times Cl_2O_2 + HCl + HOCl + ClONO_2 + BrCl$) concentrations are anomalously low (Fig. 4d). This result indicates the conversion of chlorine reservoir species to active chlorine in the air from the polar region[43]. There is also a significant negative correlation between PV anomalies and nitric acid ($HNO_3$) anomalies (Fig. 4e), suggesting that decreased $HNO_3$ concentrations within the polar vortex shifted over Eurasia may be related to the formation of type I polar stratospheric clouds (PSCs) and $HNO_3$ may be removed more strongly from the stratosphere during those years. In fact, the temperature in January during all 3 years is <195 K (Supplementary Fig. 6), close to the formation threshold of type I PSCs[44,45]. Low temperature during late winter (Fig. 4f) favours the formation of PSCs and production of active chlorine by heterogeneous chemistry on the PSCs surface within the polar vortex. The air parcels with high levels of active chlorine species are brought to the Eurasian continent due to the polar vortex shift towards Eurasia, resulting in a strong chemical ozone loss there (Fig. 4b).

**Future ozone changes associated with polar vortex shift**. Zhang et al.[38] pointed out that sea-ice loss over the Barents-Kara sea (BKS) is one of the main causes of the polar vortex shift. Furthermore, the February ENAD is found to have a significantly negative correlation with the BKS sea-ice cover in the preceding fall and winter (Supplementary Fig. 7a), indicating that the positive ENAD trend (Fig. 2d, e) may also be related to the BKS sea-ice loss (Supplementary Fig. 7b). An important and

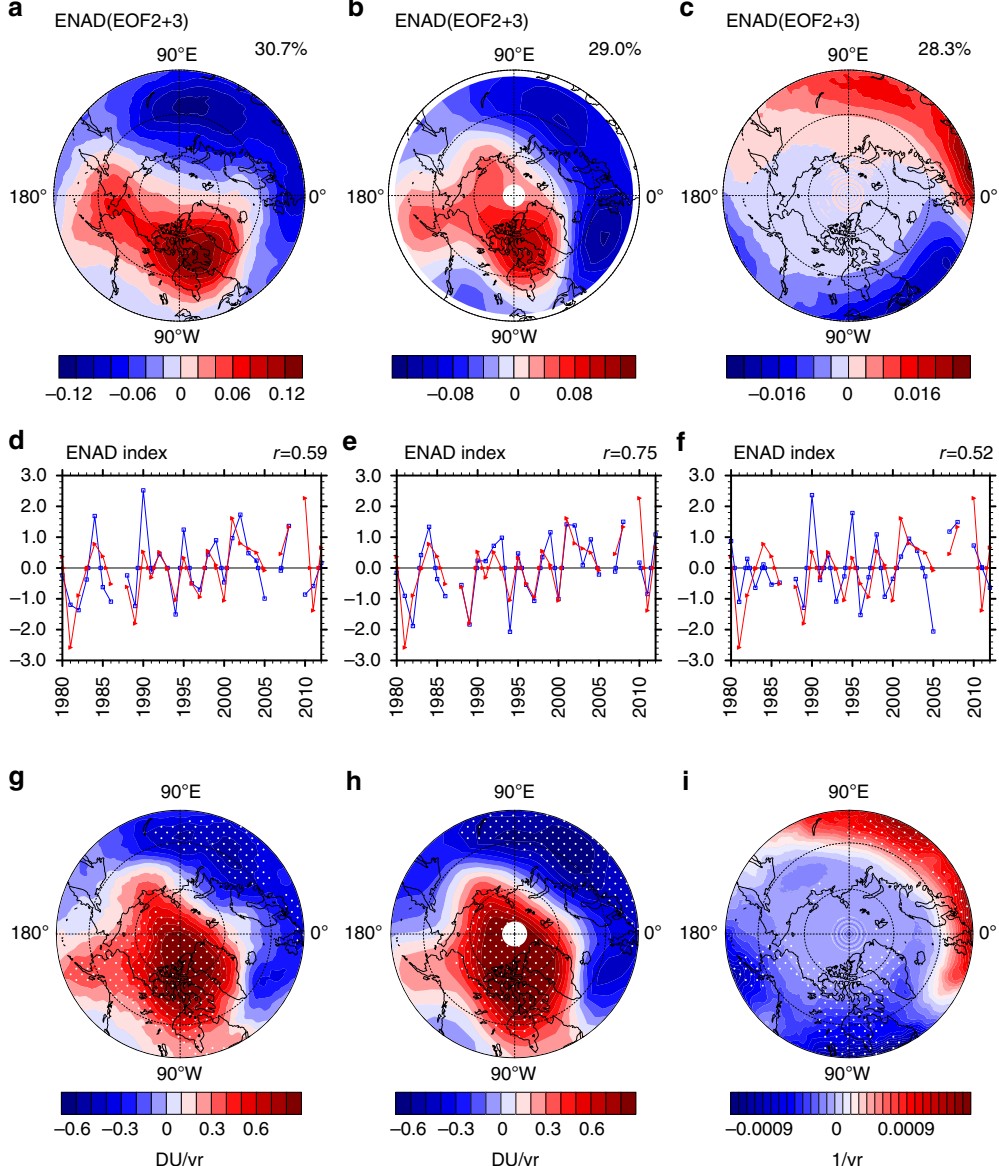

**Fig. 2** Eurasia-North America dipole mode of total column ozone and clear-sky ultraviolet radiation. **a–c** Spatial patterns of Eurasia-North America dipole (ENAD) mode (EOF2 + EOF3) and **d–f** time series of normalized ENAD index (principal component (PC)2 + PC3) (blue lines) of February mean total column ozone (TCO) over 45–90°N derived from **a, d** MSR2 data and **b, e** SLIMCAT full chemistry simulation, and **c, f** February mean clear-sky ultraviolet radiation index derived from MSR2 data. Time series of vortex shift index (see Methods section, red lines) are overlaid in **d–f**. The percentage of explained variance is shown in the top right of **a–c**, and the correlation coefficients between PC2 + PC3 and the vortex shift index are shown in the top right of captions in **d–f**. Linear trends of **g** MSR2 TCO, **h** SLIMCAT TCO and **i** MSR2 UV index regressed on the vortex shift index are also shown. The linear trends over the dotted regions are statistically significant at the 90% confidence level according to the Student's *t* test. The minimum latitude of polar stereographic projections is 45°N. The values in 1987, 2006 and 2009 are not included because the polar vortex broke up and its shape was distorted during February in these years (Methods)

interesting question is how ENAD would change in the future under the influence of continuing Arctic sea-ice loss? To address this issue, we examine historical and future scenario CCM simulations performed within the framework of the Chemistry-Climate Model Initiative phase-1 (CCMI-1)[46]. By evaluating the two leading modes of Northern Hemisphere TCO and polar vortex shift in the historical simulations (REF-C1 runs), three CCMs, i.e., IPSL, EMAC-L90MA and MRI-ESM1r1, out of a total of 13 models are identified to successfully reproduce the observed TCO variability (see Supplementary Figs. 8–10 for detailed evaluation). These CCMs also well capture a positive correlation between vortex shift index and ENAD index (Supplementary Fig. 9), in good agreement with the observation. This result

provides additional evidence that the ENAD mode is indeed related to the polar vortex shift.

Figure 5 shows the multi-model mean spatial patterns of EOF1 and ENAD modes and time series of PC1 and ENAD indices for the period 2010–2050 derived from the CCMI-1 REF-C2 (future) simulations of the three CCMs. The EOF1 and PC1 time series again reflect the overall change of TCO over the Northern Hemisphere. As in observations, the PC1 and vortex strength index are positively correlated with each other. Note that the PC1 time series in these simulations linearly decrease from 2010 to 2050 (Fig. 5a, b), this negative trend may be related to stratospheric ozone recovery in response to ODS decline and increasing green-house gases in the future[26,47].

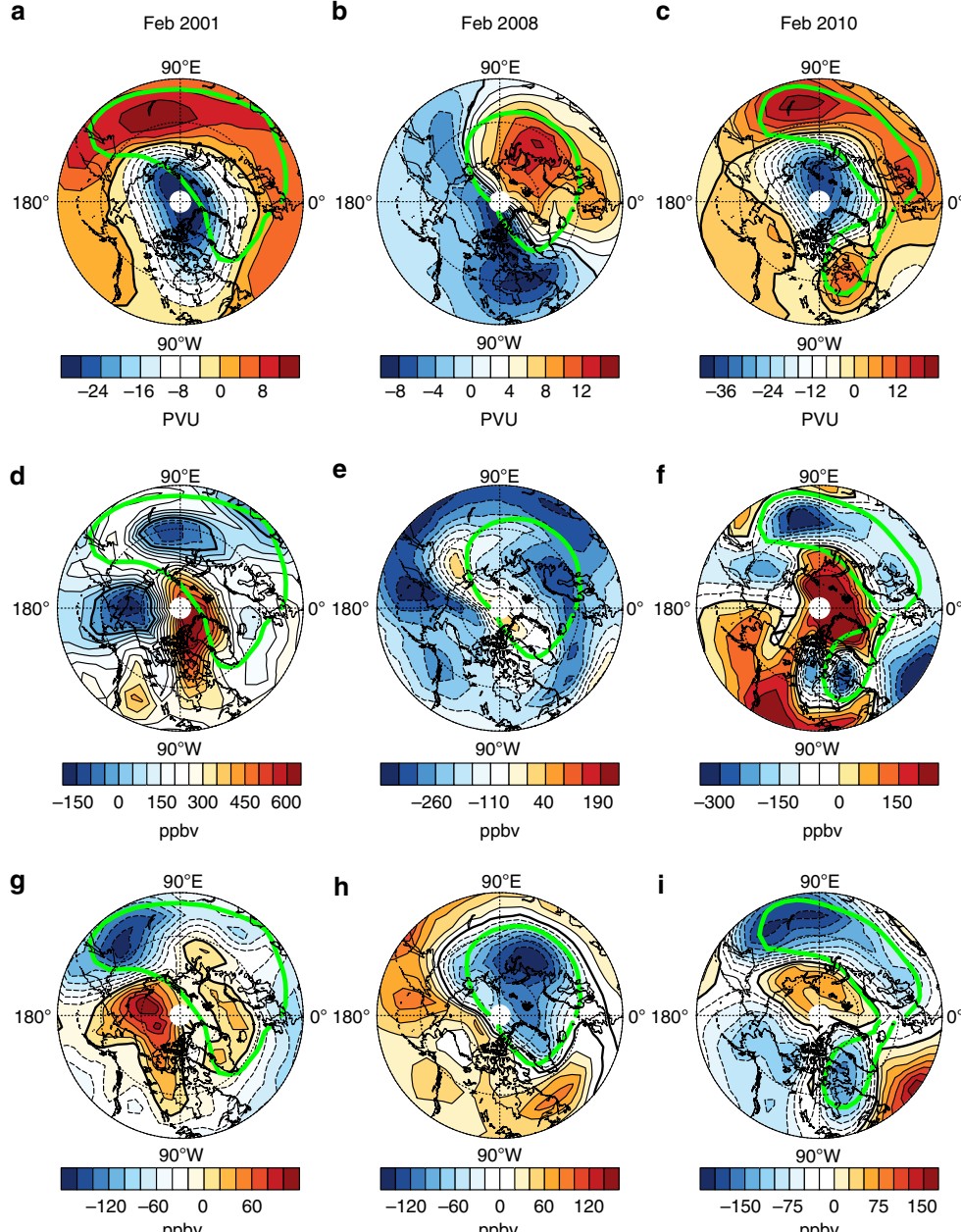

**Fig. 3** Potential vorticity and stratospheric ozone anomalies. Isentropic potential vorticity (PV) anomalies in February during **a** 2001, **b** 2008 and **c** 2010 with respect to the climatology averaged over the isentropic layers from 430 to 600 K. Panels **d**–**f** are the same as **a**–**c**, but for dynamical ozone anomalies from SLIMCAT. Panels **g**–**i** are the same as **a**–**c**, but for heterogeneous chemical ozone anomalies from SLIMCAT. The green contour lines represent the edges of the polar vortex. The minimum latitude of polar stereographic projections is 45°N

The simulated ENAD pattern shows a negative centre over the Eurasian continent and a positive one over North America as in the observations (compare Fig. 5c and Fig. 2a). There is also a significant positive correlation between ENAD index and polar vortex shift index in these future simulations (Fig. 5d). More importantly, both ENAD and the polar vortex shift index show an increasing trend for the period 2010–2041, then decreasing trend afterward. This suggests that February TCO recovery over the Eurasian continent may be continuously delayed by ENAD during the first four decades of the 21st century. Given the relationship between the polar vortex shift and sea-ice loss revealed by Zhang et al.[38], the ENAD trends may be related to different sea-ice trends during different periods. Supplementary Figure 11a indeed shows a sea-ice loss over the BKS and the polar

vortex shift increase during 2010–2041 (Fig. 5d), but a slight increase in regional sea-ice concentration (Supplementary Fig. 11b) and a decline in polar vortex shift index during 2041–2050 (Fig. 5d), presumably due to natural variability[48].

## Discussion

Our present analysis shows that the first leading mode of inter-annual variability of February TCO is associated with the polar vortex strength. In contrast, the sum of the second and third modes, referred to as the ENAD mode, is closely related to the polar vortex shift towards Eurasia, and accounts for ~30% of February TCO variance. Importantly, this mode, which has increased in amplitude during the last three decades, is associated

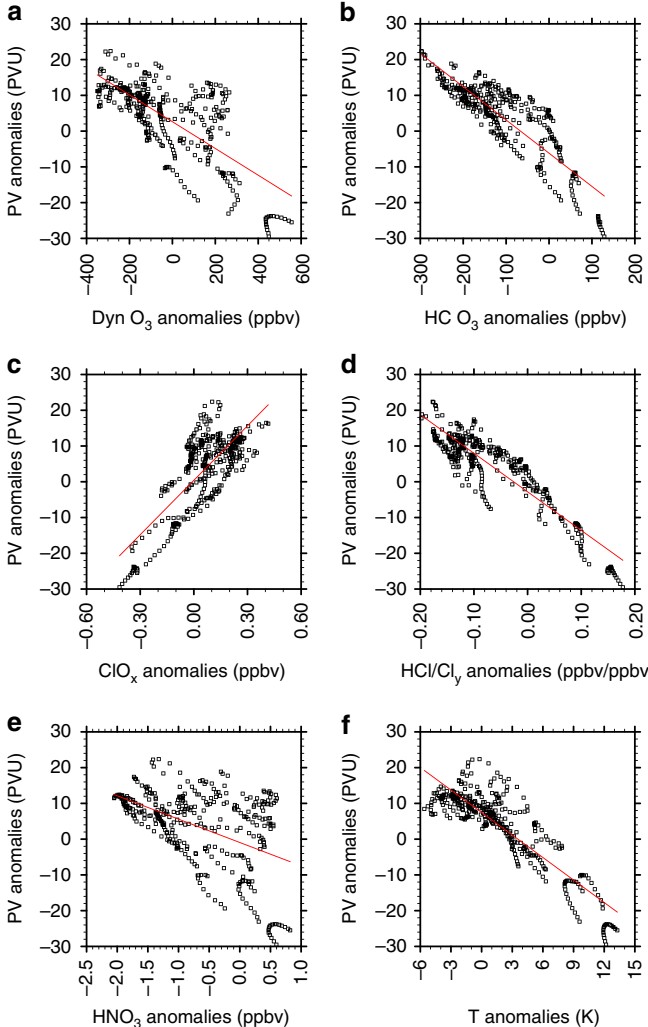

**Fig. 4** Potential vorticity anomalies versus chemical variables and temperature anomalies inside the polar vortex. Scatter plots of February potential vorticity (PV) anomalies against **a** dynamical ozone, **b** heterogeneous chemical ozone, **c** $ClO_x$, **d** $HCl/Cl_y$ ratio, **e** $HNO_3$ and **f** temperature anomalies within the polar vortex over the Eurasian continent in February during the three vortex-shift winters (i.e., 2001, 2008 and 2010). The red line represents the regression fit

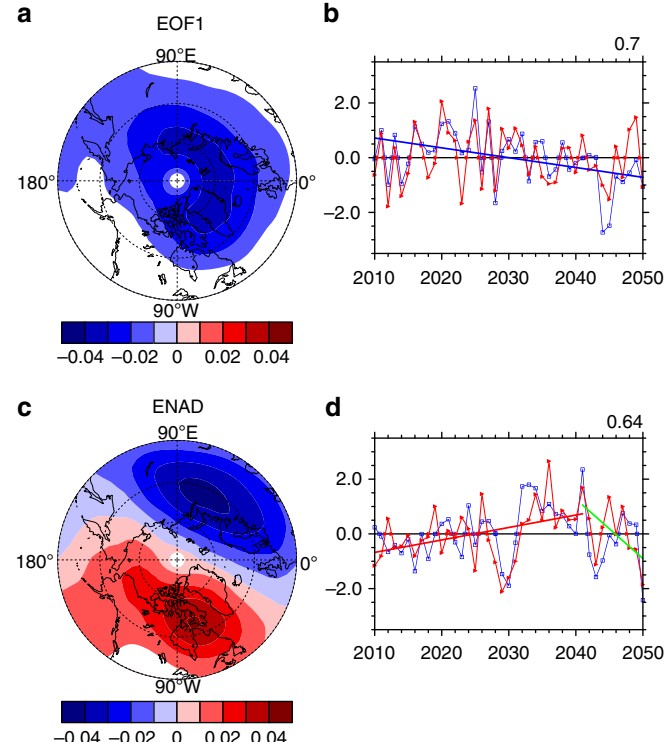

**Fig. 5** Future changes in total column ozone and polar vortex. Spatial patterns of **a** EOF1 and **c** Eurasia-North America dipole (ENAD) mode, and time series of **b** principal component (PC) 1 (blue curved line) and **d** ENAD index (blue curved line) of multi-model mean total column ozone (TCO) in February, derived from the EMAC-L90MA, MRI-ESM1r1 and IPSL models, for the period 2010–2050. The red curved lines in **b** and **d** represent the polar vortex strength and shift index, respectively. The blue straight line in **b** represents the linear trend in PC1 during 2010–2050. The red and green straight lines in **d** denote linear trends in polar vortex shift index during 2010–2041 and 2041–2050, respectively. The correlation coefficient between PC1 and the vortex strength is shown in the top right of **b**, and correlation coefficient between ENAD index and the vortex shift index is shown in the top right of **d**

with a faster TCO decline over the Eurasian continent than over the North American continent. The ENAD-like mode is also found in the clear-sky UV index over the Northern Hemisphere, suggesting that the polar vortex shift has implications for regional UV radiation changes, notably enhancing the clear-sky UV radiation level over the Eurasian continent. The SLIMCAT simulations further show that this fast Eurasian TCO decline is caused by both dynamical and chemical processes: the polar vortex shift towards Eurasian continent brings ozone-poor air within the polar vortex to Eurasia, while carrying air rich in active chlorine from the polar region to mid- and high latitudes leads to anomalously high active chlorine levels and chemical ozone loss over Eurasia.

It is well documented that, while stratospheric ozone depletion and recovery is overall driven by anthropogenic chlorine and bromine chemistry linked to ODS emissions, its rate varies significantly from region to region. The present study suggests that the polar vortex shift, partly linked to Arctic sea-ice loss in the recent past, has contributed to regional TCO decline in the Northern Hemisphere extratropics. The results from three CCMs

indicate that the polar vortex shift tends to slow down the stratospheric ozone recovery during later winter over the Eurasian continent and that this effect is likely to persist until the middle of the 21st century. Nonetheless, the ENAD trend may fluctuate and even reverse (Fig. 5d) because of internal decadal variability in BKS sea-ice cover (Supplementary Fig. 11b). Overall, the findings suggest that investigating the influence of climate change on future ozone recovery over Eurasia should not only focus on the impact of temperature changes on ozone chemistry, but also need to take account of the polar vortex changes associated with Arctic sea-ice loss.

## Methods

**Data sets**. The TCO and clear-sky UV index data for the period 1980–2012 used in this study are derived from the multi-sensor reanalysis version 2 (MSR2) data set, which assimilates data from several total ozone satellite instruments, including the total ozone mapping spectrometer, the solar backscatter UV instrument and the global ozone monitoring experiment[40]. Note that the MSR2 data set only provides public TCO data until 2012, which limits the analysis period in this study to 1980–2012. The National Institute of Water and Atmospheric Research (NIWA) assimilated ozone column data, which has a horizontal resolution of 1° latitude × 1.25° longitude and covers the time period from January in 1979 to November in 2012, is also employed. More details about the NIWA data can be found in Bodeker et al.[41] For the atmospheric dynamical fields, ERA-Interim reanalysis data sets[42] from the European Centre for Medium-Range Weather Forecasts (ECMWF) are used. The ERA-Interim TCO data are also used in this study.

**SLIMCAT model**. To simulate the stratospheric ozone distribution and its variations under a range of forcings, we use the TOMCAT/SLIMCAT off-line three-dimensional chemical transport model[39]. This model uses horizontal winds and temperatures from the ERA-Interim reanalysis data set[42]. Vertical advection is calculated from the divergence of the horizontal mass flux[39], and chemical tracers are advected conserving second-order moments[49]. Supplementary Figure 1 shows that the time series of model-calculated monthly mean of TCO anomaly averaged over the Eurasian continent is in very good agreement with that derived from the MSR2 data, giving confidence in the ability of the SLIMCAT model to reproduce the long-term variability and trend of TCO over Eurasia.

The SLIMCAT model also contains a passive odd-oxygen tracer that is set equal to the modelled chemical $O_x = O(^3P) + O(^1D) + O_3$ concentration on 1 December every year for the Northern Hemisphere and then advected passively without chemistry. At any point and time after 1 December, the difference between this passive $O_x$ and the chemically active $O_x$ represents the effect of the chemical processes on ozone (i.e., net chemical $O_x$ change)[50]. $O_x$ is mainly $O_3$ in the stratosphere where the concentrations of $O(^3P)$ and $O(^1D)$ are small, especially in winter when there is no sunlight in the polar region. In this study, the passive $O_x$ is referred to as dynamical ozone, while the chemically integrated $O_x$ is called as chemical ozone. Two numerical experiments were performed with the SLIMCAT model: the first one uses full chemistry and the second one employs full chemistry, but without HC processes. The difference in ozone between the two simulations is referred to as HC ozone.

**Polar vortex analysis**. We used the method proposed by Nash et al.[51] to define the vortex edge; i.e., the location of the largest Ertel PV gradient, with an additional constraint on the location of the maximum of the westerly jet. The years 1987, 2006, 2009 and 2013 are not included in our analysis because the polar vortex broke up early and was not well defined during February in these years due to major sudden stratospheric warming events that lasted >15 days[38]. The vortex shift index is defined as the fractional area of the geographical regions over the Eurasian continent (50–75°N, 0–120°E) covered by the polar vortex (see Fig. 1c in Zhang et al.[38]). A higher vortex shift index corresponds to a more significant shift of the polar vortex towards Eurasia.

**EOF analysis**. The EOF analysis is performed on monthly TCO data in February as a function of longitude and latitude. The grid points in the horizontal data domain (45–90°N) were weighted by the square root of the cosine of latitude. A vector time series $\mathbf{x}(t)$ represents February TCO record. The covariance matrix of $\mathbf{x}$ is given by $\mathbf{C} = \langle \mathbf{xx^T} \rangle$, where the brackets denote probabilistic expectation and the superscript $\mathbf{T}$ denotes the transpose operator of vector. The EOFs are defined as the eigenvectors $\mathbf{e}_k$ of $\mathbf{C}$ : $\mathbf{Ce}_k = \mu_k \mathbf{e}_k$, with corresponding eigenvalues $\mu_k$. The time series of TCO anomaly can be represented by $\mathbf{x}(t) = \sum_{n=1}^{N} \alpha_n(t)\mathbf{e}_n$, where the expansion coefficient time series $\alpha_n(t)$ are the principal components (PC). Overall, EOF analysis expresses the TCO series as the superposition of $N$ mutually orthogonal spatial patterns modulated by $N$ mutually uncorrelated time series. More details can be found in Monahan et al.[52]

**Code availability**. The code of TOMCAT/SLIMCAT is available through the URL: http://www.see.leeds.ac.uk/research/icas/research-themes/atmospheric-chemistry-and-aerosols/groups/atmospheric-chemistry/tomcatslimcat. Maps and plots were made with the NCAR Command Language (version 6.4.0) (software), 2017. Boulder, Colorado: UCAR/NCAR/CISL/TDD. https://doi.org/10.5065/D6WD3XH5.

**Data availability**. The original observational data are publicly available and can be downloaded from the corresponding websites (MSR2 TCO data: http://www.knmi.nl/kennis-en-datacentrum/publicatie/multi-sensor-reanalysis-of-total-ozone; http://www.temis.nl/protocols/O3global.html; NIWA TCO data: http://www.bodekerscientific.com/data/total-column-ozone; ERA-Interim data: https://www.ecmwf.int/en/forecasts/datasets/reanalysis-datasets/era-interim; Hadley Centre Sea Ice data set: https://www.metoffice.gov.uk/hadobs/hadisst). The CCMI-1 model simulations can be downloaded from: http://blogs.reading.ac.uk/ccmi/badc-data-access.

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

## Acknowledgements

Funding for this project was provided by the National Natural Science Foundation of China (41630421, 41575039, 41705022 and 41575038). J.Z. is supported by the Fundamental Research Funds for the Central Universities (lzujbky-2017-4). S.-W.S. was partly supported by 'Climate Change Correspondence Program' of Korea Ministry of Environment. We acknowledge the data sets from the MSR2 and ECMWF. We would like to thank Dr. Greg Bodeker of Bodeker Scientific, funded by the New Zealand Deep South National Science Challenge, for providing the combined TCO (NIWA ozone) database. The SLIMCAT modelling work was supported by the UK National Centre for Atmospheric Science (NCAS). W.F. acknowledges the support of NCAS fund. We also acknowledge the CCM modelling groups for making their simulations available for this analysis, the joint WCRP SPARC/IGAC Chemistry-Climate Model Initiative (CCMI) for organizing and coordinating the model data analysis activity, and the British Atmospheric Data Centre (BADC) for collecting and archiving the CCMI model output. N.L.A. and A.T.A. acknowledge that their work used the ARCHER UK National Supercomputing Service (http://www.archer.ac.uk) and the MONSooN system, a collaborative facility supplied under the Joint Weather and Climate Research Programme, which is a strategic partnership between the UK Met Office, and N.B. and F.M.C. were supported by the Joint UK BEIS/Defra Hadley Centre Climate Programm (GA01101). N.B. and S.B. were also supported by the European Commission's Seventh Framework Programme, under grant agreement no. 603557, StratoClim project. S.B. acknowledges funding by the LABEX L-IPSL project (grant ANR-10- LABX-18-01). The EMAC simulations have been performed at the German Climate Computing Centre (DKRZ) through support from the Bundesministerium für Bildung und Forschung (BMBF), Germany. DKRZ and its scientific steering committee are gratefully acknowledged for providing the HPC and data archiving resources for this consortial project ESCiMo (Earth System Chemistry integrated Modelling). O.M. and G.Z. acknowledge the UK Met Office for use of the MetUM. Their research was supported by the NZ Government's Strategic Science Investment Fund (SSIF) through the NIWA programme CACV. O.M. acknowledges funding by the New Zealand Royal Society Marsden Fund (grant 12-NIW-006) and by the Deep South National Science Challenge (http://www.deepsouthchallenge.co.nz). O.M. and G.Z. also wish to acknowledge the contribution of NeSI high-performance computing facilities to the results of this research. New Zealand's national facilities are provided by the New Zealand eScience Infrastructure (NeSI) and funded jointly by NeSI's collaborator institutions and through the Ministry of Business, Innovation and Employment's Research Infrastructure programme (https://www.nesi.org.nz).

## Author contributions

J.Z., W.T. and F.X. contributed to the paper writing, design of numerical experiments and data analysis. M.P.C., W.F. and S.-W.S. contributed to the discussion and paper writing. M.P.C., N.L.A., A.T.A., S.B., N.B., M.D., S.D., P.J., D.K., O.K., M.M., O.M., F.M.O'C., G.P., D.A.P., L.E.R., E.R., D.V. and G.Z. contributed to the CCMI simulations. W.W. and Y.H. contributed to the data analysis. All authors reviewed the manuscript.

## Additional information

**Competing interests:** The authors declare no competing financial interests.

