## [Peer Review File · Nature Communications]

Reviewers' comments:

Reviewer #1 (Remarks to the Author):

The authors study the correlation between the variation of total column ozone (TCO) and the deformation of the Arctic polar vortex over the past thirty years. By using empirical orthogonal function analysis (EOF), they find a new mode which explains the 30 % variance in TCO and it correlates well with the displacement of the polar vortex towards Eurasia. In light of this, they claim that despite the overall general reduction of ozone-depleting substances after the Montreal Protocol, the expected increase in stratospheric ozone concentration in the coming decades can be spatially inhomogeneous and be significantly influenced by the deformation of the polar vortex. I think that the paper is suitable for publication and well written. I believe, however, that addressing the following comments would improve the quality and the rigorousness of the manuscript.

COMMENTS

1. Line 42. The way to address the polar vortex strength should be briefly mentioned in the main text and not only in the caption of the supplementary Fig. 2 (a reference would also work). From supplementary Fig. 2, I see that the authors quantify the polar vortex strength from averaged geopotential height over a 50 hPa isobaric surface. I have the following questions.

(a) Is this a common method to address the polar vortex strength? If so, please insert a reference.

(b) Under certain assumptions, the 3D polar vortex is approximately 2D on isentropic surfaces. Hence I suggest this would be a better choice instead of the isobaric surfaces. Furthermore, this would allow consistency with other analysis the authors did using potential vorticity (PV) and other quantities given on isentropic surfaces.

(c) Why do the authors use only the geopotential height on 50 hPa isobaric surface, instead of averaging the same quantity over multiple isobaric surfaces intersecting the polar vortex?

(d) How would the results change if, for example, the strength of the polar vortex is measured by averaging PV over isentropic surfaces rather than geopotential height over isobaric surfaces?

2. Line 59. Please insert "of Figure 1" after "left panel" and "right panel".

3. Line 60. The authors claim that over the last 30 years, the polar vortex always experienced significant changes only in the month of February. This is somehow surprising to me. For example, in the early January 2014, due to the deformation of the arctic polar vortex, an exceptional cold weather has been recorded in the northeastern US. Perhaps using yearly averaged data of December, January and February, instead of only February, the authors may check the robustness of their results.

4. General comment. Despite commonly used in the atmospheric and oceanic communities, PV is not suitable for material assessment of fluid flows. Firstly, it is not frame invariant, and hence the definition of the vortex edge would change if the same phenomenon would be observed by different observers moving relative to each other. This violates a basic requirement for a self-consistent material assessment. Examples of material features include the existence, displacement and deformation of a vortex. Secondly, assessing the vortex edge using PV-based methods would identify non-material structures, and hence any conclusion related to transport within the vortex would be unjustified. Even in the ideal case in which PV is conserved, the PV contours which have the largest PV gradient with respect to equivalent latitude (i.e., the Nash method) at two distinct times, do not

advect into each others. The authors may look here: <https://arxiv.org/abs/1702.05593> for further details on this. Having said that, there is a correlation between the vortex edge and PV. I expect that the results would not change substantially, but may further improve if the polar vortex edge is identified by more rigorous methods.

5. Lines 122-123. I don't understand the last sentence.

Reviewer #2 (Remarks to the Author):

The manuscript of Zhang et al., presents an analysis of past stratospheric ozone changes for the Northern Hemisphere (NH) using a three-dimensional (3D) chemical transport model (CTM), global meteorological re-analyses and satellite total ozone datasets and statistical methods. It is based on recent previous work by the same authors that established that the late winter NH polar vortex has shifted persistently towards the Eurasian continent over the past three decades, associated with Arctic sea-ice loss induced by global warming. The current work reveals a very important implication of this shift, related to a dipole spatial structure in the inter-annual variability and trend of the NH total ozone column (TOC) in the latitudes northern than 45 degrees. It does so in an unequivocal manner, using eigen-value statistical computations to demonstrate the existence of a Eurasia-North America Dipole (ENAD) pattern in the TCO observed and modelled fields. It is also shown that the TCO decline over Eurasia, linked to the Arctic vortex shift, is brought about by dynamical and chemical ozone loss factors. This is made possible via the SLIMCAT CTM set-up and specialised simulations that can distinguish between "dynamical" and "chemical" ozone as well as for gas-phase only and heterogeneous ozone depletion.

The manuscript is nicely written, presentation of results is appealing and the underlying science and methods are sound. This work contains novel results which highlight the important connection between climate change and stratospheric ozone and the conclusions reveal its regional manifestation, also in the crucial context of climate change effects to future ozone hole recovery. The topic should be of interest to fields other than atmospheric chemistry and climate change, for example human health or biodiversity (for the possible implications on surface UV). I recommend publication once the following minor points are dealt with (referring to the manuscript with the corresponding lines).

1) line 28: Some more credit to related ground-breaking or more recent previous work on the long-term dynamical and chemical ozone changes is needed, by referring to the papers by Fusco and Salby (1999) and Harris et al. (2008).

2) lines 45-46: You might want to add in half-sentence the global warming induced sea-ice loss relation to the polar vortex shift from the previous Zhang et al work, especially since you connect it to climate change in the following sentence, but without specifying the link.

3) line 117: replace "vortexes" with "vortices"

4) lines 125-127: A brief 1-2 lines sentence is needed here to define the "dynamical and chemical ozone" before you start using it. I had to interrupt the reading and look for the detailed explanation in the Methods (in SLIMCAT model) so a brief definition at the beginning of this paragraph will help.

REFERENCES

Fusco, A., and M. L. Salby, (1999) Interannual variations of total ozone and their relationship to

variations of planetary wave activity. *J. Climate*, 12, 1619–1629.

Harris, N., et al. (2008), Ozone trends at northern mid- and high latitudes—A European perspective, *Ann. Geophys.*, 26, 1207–1220, doi:10.5194/angeo-26-1207-2008.

Reviewer #3 (Remarks to the Author):

It is not at all surprising or novel that ozone concentrations co-vary with the location of the polar vortex (since the polar vortex acts as a containment vessel and facilitates ozone depletion).

In my view, the authors would have done better to combine the key results of this paper with those of Zhang et al. 2016, creating one coherent and consistent story. This currently feels like an attempt to extract two separate manuscripts from the same story.

The introduction lacks some basic introduction to the dynamical differences between the Arctic and Antarctic polar vortices. Some of the factors alluded to arise simply from the fact that wave driving is larger in the Northern Hemisphere, making the polar vortex more perturbed than it is in the SH. The Arctic vortex is more disturbed on both intraseasonal and interannual timescales, whereas the Antarctic vortex is climatologically stronger, allowing for substantial ozone depletion. (e.g, Waugh and Polvani 2010; Solomon et al. 2014; Sheshadri et al. 2015 and probably many others, this is now almost textbook material).

In looking at the polar vortex shift, did you make allowances for the fact that in about 6 years out of 10, the polar vortex underwent major warming events? Particularly when there is a “split” event, the vortex splits into two daughter vortices, the bigger one of which is over Eurasia and the smaller one of which is over North America (Matthewman et al. 2008). Perhaps what you are seeing as a trend towards a shift in the location of the vortex is simply averaging over these events? There were 12 subsequent years with SSWs starting in the early 2000's.

L23 Most such studies have focused on the SH, where ozone depletion has been substantial. Ozone anomalies or trends have been quite small in the NH in comparison.

L47 EOF analysis of what? Daily ozone concentrations as a function of longitude and latitude? Was the variable of interest weighted in any way?

L62 There is some longitudinal structure

L117 you mean vortices

L126 it is not active “transport” in the way that this sentence implies. The “balanced” state is that that vortex itself is located in a different position, and therefore the ozone concentrations appear shifted in location. “Transport” implies transport by the circulation.

Reviewer #4 (Remarks to the Author):

This study can be considered as the “part 2” of a recent paper published by the same group of authors in *Nature Clim. Change* (Zhang et al. 2016). Therefore, its background mainly relies on this previous “part 1” investigation. In short, sea ice depletion over the North Pole coupled with an increased snow precipitation over the north Asia worked together in shifting the polar vortex over the Eurasian continent and away from North America during the last decades. With this in mind, the new study focuses on the ozone changes induced by this shift. By analyzing ozone reanalysis and simulations

carried out with SLIMCAT model the authors attribute these changes to transport of ozone-poor air from Arctic and to local chemical processes caused by the transport of active chlorine species towards lower latitudes.

Overall, the study is well planned, results are convincing and it provides a useful contribution to evaluate the ozone temporal evolution/recovery at northern high latitudes. Nevertheless, in the present form this work can hardly be published in a high ranking journal while I find it more suitable for JGR or ACP. Personally, I would find more convenient having included some of the present findings within the part 1, but I understand the need of splitting the work, so I don't blame the authors. However, having already achieved the main results in the part 1, the additional findings presented in this new paper look somewhat ordinary and far from being very intriguing. The authors state that "The question remains whether this shift of the polar vortex induced by climate change can change trace gas distributions, especially the ozone distribution in the Northern Hemisphere during late winter". In the light of the part 1 of the study, it seems rather plausible that ozone would follow the vortex shift so one can expect what the authors described here. Obviously, the study has some merit e.g. the assessment of the ozone transport with respect to the local chemical processes, but I think that at the present stage this is not enough to warrant publication on Nature communication and some further work should be carried out. Also, the conclusions of this study don't look novel (e.g. "The results suggest that the future ozone recovery in late-winter would be sensitive not only to ODSs change but also to the polar vortex change") and I believe that a broader paper instead of a short communication could better address the issues here mentioned.

I actually don't have specific suggestions on how making this work more appealing, nevertheless I still see that the analysis is basically limited to SLIMCAT and MSR2 datasets. Due to the limited temporal extension of the time series and the expected large internal climate variability in the northern high latitudes, can an additional analysis based on climate models (either from the CCMI or CMIP5) be useful in this context? Moreover, since things seem to be somewhat clear for the "historical" period, I would find even more remarkable the investigation of the future ozone response under different RCP scenarios. Then, is MSR2 the best observational reference for this study? You should justify this choice. Recently, many combined O3 datasets spanning more than three decades are become available. Although they mostly rely on solar occultation instruments and therefore are characterized by a sparse and infrequent sampling at high latitudes, the new SBUV v8.6 datasets could still be useful. Moreover, since both the data assimilation of MSR2 as well as the SLIMCAT model are driven by ERA-interim, it could be interesting including an additional independent assimilated ozone dataset. A good candidate can be the Bodeker's NIWA ozone dataset. Finally, as ERA-interim assimilates also ozone fields, perhaps, despite its limitations, ERA-interim ozone could even be useful for checking the consistency of these results.

We thank the four reviewers very much for their important comments. This study, for the first time, highlights the environmental and chemical influences induced by the polar vortex shift revealed by Zhang et al. (2016). In the original version of this paper, we mainly analyzed the historical influence of the polar vortex shift on the ENAD mode of TCO over the Northern Hemisphere (NH) and proposed that the ENAD could delay the ozone recovery over Eurasia in the future. In the revision, we found the NH clear-sky ultraviolet variation is closely related to ENAD mode. Furthermore, we make more efforts to verify that the ENAD would be significantly enhanced in the near future by using results from Chemistry-Climate Model Initiative (CCMI) experiments. The revised manuscript is substantially improved and we have made the following major changes in response to the reviewers' comments and suggestions:

- 1) We used several methods to calculate polar vortex strength and shift indices. The results indicate that our analysis is not sensitive to the method used for determining the vortex edge.**
- 2) To highlight the importance of ENAD in the future, we analyzed future ENAD changes for the period 2010-2050 using the simulations from CCMI. It is found that the polar vortex shift index shows an increasing trend during 2010-2041 while it decreases after 2041. This trend variation may be related to different sea-ice trends in different periods, i.e., there is still a negative trend in sea-ice over the Barents and Kara seas and the polar vortex still shows a tendency to shift towards Eurasia during 2010-2041. By contrast, the sea-ice shows an increasing trend after 2041, corresponding to a decreasing trend in polar vortex shift index during 2041-2050. This analysis suggests that the positive ENAD trend likely continues in the coming decades, slowing down the stratospheric ozone recovery during late winter over Eurasia in near future 30 years.**
- 3) Some parts of the introduction have been rephrased in order to highlight the significance of Arctic ozone research and outstanding scientific questions related to this topic.**

Our detailed replies are given below.

Responses to Referee 1

The authors study the correlation between the variation of total column ozone (TCO) and the deformation of the Arctic polar vortex over the past thirty years. By using empirical orthogonal function analysis (EOF), they find a new mode which explains the 30% variance in TCO and it correlates well with the displacement of the polar vortex towards Eurasia. In light of this, they claim that despite the overall general reduction of ozone-depleting substances after the Montreal Protocol, the expected increase in stratospheric ozone concentration in the coming decades can be spatially inhomogeneous and be significantly influenced by the deformation of the polar vortex. I think that the paper is suitable for publication and well written. I believe, however, that addressing the following comments would improve the quality and the rigorousness of the manuscript.

Response: We thank the reviewer for his/her comments. We have revised the manuscript carefully according to these comments and suggestions. The detailed point-to-point responses are as follows.

Comments:

1. Line 42. The way to address the polar vortex strength should be briefly mentioned in the main text and not only in the caption of the supplementary Fig. 2 (a reference would also work). From supplementary Fig. 2, I see that the authors quantify the polar vortex strength from averaged geopotential height over a 50 hPa isobaric surface. I have the following questions.

(a) Is this a common method to address the polar vortex strength? If so, please insert a reference.

(b) Under certain assumptions, the 3D polar vortex is approximately 2D on isentropic surfaces. Hence I suggest this would be a better choice instead of the isobaric surfaces. Furthermore, this would allow consistency with other analysis the authors did using potential vorticity (PV) and other quantities given on isentropic surfaces.

(c) Why do the authors use only the geopotential height on 50 hPa isobaric surface, instead of averaging the same quantity over multiple isobaric surfaces intersecting the polar vortex?

(d) How would the results change if, for example, the strength of the polar vortex is

measured by averaging PV over isentropic surfaces rather than geopotential height over isobaric surfaces?

Response: Thanks for your comments and suggestions. The method to calculate polar vortex strength has been explicitly introduced in the main text (L74-L75) and the related four questions are replied as follows:

Following reviewer's comments, the sensitivity of the vortex strength to the details of methods is tested and summarized in Figure R1. Fig. R1 shows normalized time series of the polar vortex strength derived from four different methods, i.e., 50 hPa geopotential height (Kolstad et al., 2010), the first empirical orthogonal function (EOF) of zonal wind at 50 hPa (Limpasuvan et al., 2004), Northern Annular Mode (NAM) index (Baldwin et al., 2001) and PV averaged between the layer 430-600 K. Note that the time series of the polar vortex strength derived from different methods are well consistent with each other, suggesting that the time variations of the strength of the stratospheric polar vortex are not sensitive to its definition. According to the reviewer's suggestion, we choose the PV averaged between 430 and 600 K as the strength of polar vortex in the revised paper for the purpose of keeping consistency with other analysis on the isentropic surfaces.

Figure R1. Normalized time series of stratospheric vortex strength calculated based on (blue) 50 hPa geopotential height, (black) the EOF1 of zonal wind at 50 hPa, (green) NAM index at 50 hPa, and (red) PV averaged between the layer 430-600 K. The NAM index is calculated for each pressure level by projecting daily geopotential height onto the EOF1 pattern

(Baldwin et al., 2001). All indices are averaged between 65-90°N, except that the NAM index is the EOF1 of 90-day low-pass filtered geopotential anomalies north of 20°N (Baldwin et al., 2001).

In addition, the correlation coefficient between the new vortex strength index time series and PC1 of TCO to the northward of 45°N is 0.7, and the correlation coefficient between 50 hPa geopotential height and PC1 in the original manuscript is also 0.7.

References:

- Baldwin, M. P., & Dunkerton, T. J. (2001). Stratospheric Harbingers of Anomalous Weather Regimes. *Science*, 294(5542), 581-584.
- Kolstad, E. W., Breiteig, T., & Scaife, A. A. (2010). The association between stratospheric weak polar vortex events and cold air outbreaks in the Northern Hemisphere. *Quarterly Journal of the Royal Meteorological Society*, 136(649), 886-893.
- Limpasuvan, V., Thompson, D. W., & Hartmann, D. L. (2004). The Life Cycle of the Northern Hemisphere Sudden Stratospheric Warmings. *Journal of Climate*, 17(13), 2584-2596.

2. Line 59. Please insert “of Figure 1” after “left panel” and “right panel”.

Response: Corrected, thank you.

3. Line 60. The authors claim that over the last 30 years, the polar vortex always experienced significant changes only in the month of February. This is somehow surprising to me. For example, in the early January 2014, due to the deformation of the Arctic polar vortex, an exceptional cold weather has been recorded in the northeastern US. Perhaps using yearly averaged data of December, January and February, instead of only February, the authors may check the robustness of their results.

Response: We agree with the reviewer that the sentence ‘significant polar vortex change only occurs in February’ is unclear. This sentence was intended to refer to that the trend of stratospheric polar vortex shifting towards Eurasia is significant in February but not so significant in other winter months (Zhang et al., 2016). Figure R2 shows time series of polar vortex shift index from December to February. The trends of polar vortex shift in December and January are not

significant as that in February. The significant polar vortex shift only occurs in February more likely due to more upward planetary waves associated with Arctic sea ice loss in February (Kim et al., 2014; Peings et al., 2014; Zhang et al., 2016). We have rephrased this sentence in the revised paper as follows:

“Only February is considered here because the significant polar vortex shift towards Eurasia occurred mostly in this month.”

Figure R2. Time series of normalized polar vortex shift index (see Method section) and its linear trend (straight line) in December (blue), January (green) and February (red) for the period 1980-2015. Solid straight line represents the corresponding linear trends which are statistically significant at the 95% confidence levels, respectively; whereas, the linear trends denoted by the dashed lines are not significant at the 90% confidence level.

References:

Kim, B. M. et al. Weakening of the stratospheric polar vortex by Arctic sea-ice loss. Nat

Commun 5 (2014).

Peings, Y. &Magnusdottir, G. Response of the Wintertime Northern Hemisphere Atmospheric Circulation to Current and Projected Arctic Sea Ice Decline: A Numerical Study with CAM5. *J Climate* 27, 244-264 (2014).

Zhang, J., Tian, W., Chipperfield, M. P., Xie, F. & Huang, J. Persistent shift of the Arctic polar vortex towards the Eurasian continent in recent decades. *Nature Clim. Change*6, 1094-1099, doi:10.1038/nclimate3136 (2016).

4. General comment. Despite commonly used in the atmospheric and oceanic communities, PV is not suitable for material assessment of fluid flows. Firstly, it is not frame invariant, and hence the definition of the vortex edge would change if the same phenomenon would be observed by different observers moving relative to each other. This violates a basic requirement for a self-consistent material assessment. Examples of material features include the existence, displacement and deformation of a vortex. Secondly, assessing the vortex edge using PV-based methods would identify non-material structures, and hence any conclusion related to transport within the vortex would be unjustified. Even in the ideal case in which PV is conserved, the PV contours which have the largest PV gradient with respect to equivalent latitude (i.e., the Nash method) at two distinct times, do not advect into each others. The authors may look here: <https://arxiv.org/abs/1702.05593> for further details on this. Having said that, there is a correlation between the vortex edge and PV. I expect that the results would not change substantially, but may further improve if the polar vortex edge is identified by more rigorous methods.

Response: Thank you very much for your suggestion. We read through the suggested paper (Serra et al., 2017) which was useful. We tried our best to recalculate the vortex edge according to the LCSs method described in the paper. We downloaded the LCSs code from the author's website but we failed to drive it using our initial data to calculate the polar vortex edge. Instead, we used two other methods without using PV to examine the polar vortex edge according to Nash criteria. First, we recalculated the polar vortex edge as the location of maximum zonal wind with respect to equivalent latitude and show the new polar vortex area index in Figure R3 (green line). Different from the Nash method using maximum PV gradient as the polar vortex, this method is mainly based on the maximum zonal westerly as the transport barrier between mid-latitudes and

the Arctic region. It can be found the vortex area index using the criteria of maximum zonal wind has a good correlation with that using Nash criteria (blue line), with a correlation coefficient of 0.87.

In addition, we used the subpolar minimum in the probability distribution function (PDF) of the long-lived tracer N_2O in the lower stratosphere to locate the polar vortex edge (Sparling, 2000; Palazzi et al., 2011). Palazzi et al. (2011) pointed out that tracer PDFs are an equivalent “measure” of tracer gradients and the minimum values of the PDFs occur where the tracer gradient is maximum, indicating the location of a transport barrier (i.e., polar vortex edge). The conservative property of N_2O is better than PV since N_2O has long lifetime of over 10 years in the lower stratosphere and can be considered as a passive tracer (Brasseur and Solomon, 1984). We used Microwave Limb Sounder (MLS) satellite data during 2005-2015 to calculate the polar vortex edge and found that the correlation coefficient between the vortex area index inferred from N_2O PDFs (red line) and from PV gradient reaches 0.98, suggesting that the PV-based method is also reliable.

Figure R3. Time series of polar vortex area index calculated based on Nash method (blue), the maximum zonal wind method (green) and the minimum of N_2O PDFs derived from MLS data (red). Note that MLS data is only available for the period 2005-2015.

References:

- Brasseur, G., & Solomon, S. (1984). Aeronomy of the middle atmosphere: chemistry and physics of the stratosphere and mesosphere. Dordrecht D.reidel Publishing Co.p, 3.**
- Palazzi, E., Fierli, F., Stiller, G. P., & Urban, J. (2011). Probability density functions of long-lived tracer observations from satellite in the subtropical barrier region: data intercomparison. Atmospheric Chemistry and Physics, 11(20), 10579-10598.**
- Sparling, L. C. (2000).Statistical perspectives on stratospheric transport. Reviews of Geophysics, 38(3), 417-436.**

5. Lines 122-123. I don't understand the last sentence.

Response: This sentence has been rephrased to “*Correspondingly, the ENAD, which is obtained by EOF2 plus EOF3, essentially reflects the influence of polar vortex shifting towards the Eurasian continent (0-120°E sector).*”

Responses to Referee 2

The manuscript of Zhang et al., presents an analysis of past stratospheric ozone changes for the Northern Hemisphere (NH) using a three-dimensional (3D) chemical transport model (CTM), global meteorological re-analyses and satellite total ozone datasets and statistical methods. It is based on recent previous work by the same authors that established that the late winter NH polar vortex has shifted persistently towards the Eurasian continent over the past three decades, associated with Arctic sea-ice loss induced by global warming. The current work reveals a very important implication of this shift, related to a dipole spatial structure in the inter-annual variability and trend of the NH total ozone column (TOC) in the latitudes northern than 45 degrees. It does so in an unequivocal manner, using eigen-value statistical computations to demonstrate the existence of a Eurasia-North America Dipole (ENAD) pattern in the TCO observed and modelled fields. It is also shown that the TCO decline over Eurasia, linked to the Arctic vortex shift, is brought about by dynamical and chemical ozone loss factors. This is made possible via the SLIMCAT CTM set-up and specialised simulations that can distinguish between “dynamical” and “chemical” ozone as well as for gas-phase only and heterogeneous ozone depletion.

The manuscript is nicely written, presentation of results is appealing and the underlying science and methods are sound. This work contains novel results which highlight the important connection between climate change and stratospheric ozone and the conclusions reveal its regional manifestation, also in the crucial context of climate change effects to future ozone hole recovery. The topic should be of interest to fields other than atmospheric chemistry and climate change, for example human health or biodiversity (for the possible implications on surface UV). I recommend publication once the following minor points are dealt with (referring to the manuscript with the corresponding lines).

Response: We thank the reviewer for the positive evaluation of our study and we appreciate the reviewer’s very helpful comments. We have revised the manuscript carefully according to his/her suggestions.

1) line 28: Some more credit to related ground-breaking or more recent previous work on the long-term dynamical and chemical ozone changes is needed, by referring to the

papers by Fusco and Salby (1999) and Harris et al. (2008).

Response: We have added more references related to the long-term dynamical and chemical ozone changes in the revised paper as follows:

Fusco, A. C., & Salby, M. L. (1999). Interannual Variations of Total Ozone and Their Relationship to Variations of Planetary Wave Activity. *Journal of Climate*, 12(6), 1619-1629.

Andersen, S. B., & Knudsen, B. M. (2002). The influence of vortex ozone depletion on Arctic ozone trends. *Geophysical Research Letters*, 29(21), 9-1-9-4.

Rex, M., Salawitch, R. J., Der Gathen, P. V., Harris, N. R., Chipperfield, M. P., & Naujokat, B. (2004). Arctic ozone loss and climate change. *Geophysical Research Letters*, 31(4).

Rex, M., et al. (2006). Arctic winter 2005: Implications for stratospheric ozone loss and climate change. *Geophysical Research Letters*, 33(23).

Harris, N. R., et al. (2008). Ozone trends at northern mid- and high latitudes – a European perspective. *Annales Geophysicae*, 26(5), 1207-1220.

Rieder, H. E., & Polvani, L. M. (2013). Are recent Arctic ozone losses caused by increasing greenhouse gases. *Geophysical Research Letters*, 40(16), 4437-4441.

Bednarz, E. M., Maycock, A. C., Abraham, N. L., Braesicke, P., Dessens, O., & Pyle, J. A. (2016). Future Arctic ozone recovery: the importance of chemistry and dynamics. *Atmospheric Chemistry and Physics*, 16(18), 12159-12176.

2) lines 45-46: You might want to add in half-sentence the global warming induced sea-ice loss relation to the polar vortex shift from the previous Zhang et al work, especially since you connect it to climate change in the following sentence, but without specifying the link.

Response: This is a good suggestion and we have rephrased this sentence in the revised paper (please see Lines 53-55) as follows:

“Zhang et al.⁴⁰ recently reported that the late-winter Arctic vortex has shifted persistently towards the Eurasian continent over the past three decades partly due to the Arctic sea-ice loss”

3) line 117: replace “vortexes” with “vortices”

Response: Corrected, thank you.

4) lines 125-127: A brief 1-2 lines sentence is needed here to define the “dynamical and chemical ozone” before you start using it. I had to interrupt the reading and look for the detailed explanation in the Methods (in SLIMCAT model) so a brief definition

at the beginning of this paragraph will help.

Response: We have added some SLIMCAT introductions in the revised paper (please see L143-147) as follows:

“In the SLIMCAT simulation, the dynamical ozone concentrations can be calculated from a 'passive ozone tracer' advected by the model dynamical processes, while the chemical ozone depletion is represented by the difference between the modeled ozone concentrations with full chemistry and dynamical ozone concentrations (see Methods section).”

Responses to Referee 3

It is not at all surprising or novel that ozone concentrations co-vary with the location of the polar vortex (since the polar vortex acts as a containment vessel and facilitates ozone depletion).

Response: We thank the reviewer for his/her review comments. Although previous studies have revealed that the ozone concentrations co-vary with the polar vortex location in some individual cases, we, for the first time, identified a new mode of ozone anomalies in the Northern Hemisphere (NH), i.e., ENAD, which is found to significantly enhance the ozone depletion over the Eurasian Continent in the past three decades. This mode has implications in understanding long-term variations of ozone in the past and estimating the future ozone trend at NH mid- and high latitudes. In the response file, we further provide some evidence from CCMI experiments that the ENAD could likely delay the future ozone recovery.

The detailed point-to-point responses to the reviewer's comments are listed as follows.

In my view, the authors would have done better to combine the key results of this paper with those of Zhang et al. 2016, creating one coherent and consistent story. This currently feels like an attempt to extract two separate manuscripts from the same story.

Response: Thanks for your comments. This study is indeed connected with Zhang et al. (2016). However, this study focused on the long-term trend of ozone at NH mid- and high latitudes which has large uncertainties. Zhang et al. (2016) focused on the change of polar vortex location and its underlying dynamic mechanism; however, this study emphasizes the environmental and chemical influences brought by the polar vortex shift and focuses on the long-term trend of ozone at NH mid- and high latitudes which has large uncertainties. These two studies are related but have different issues.

In revised paper, we further distinguish this difference by providing more

evidence that the polar vortex shift could have an important influence on the future ozone variations at NH mid- and high latitudes.

First, we checked the performance of the models in Chemistry-Climate Model Initiative Phase1 (CCMI-1) REF-C1 experiments in simulating the two leading modes of Northern Hemisphere TCO and polar vortex shift in historical period for the purpose of picking out suitable CCMs to simulate future ENAD changes. To evaluate the performance of 13 CCMs in simulating historical spatial patterns of EOF1 and ENAD, Taylor diagram (Figure R4) is plotted, which shows how closely a spatial pattern from models matches that from observation. With regard to EOF1 of the TCO (Fig. R4a), the spatial patterns in five CCM simulations (i.e., IPSL, EMAC-L90MA, MRI-ESM1r1, NIWA-UKCA and CNRM-CM5-3) are the closest to the observations (REF in Fig. R4a). For the ENAD mode (Fig. R4b), the spatial patterns in the five simulations still keep high correlations with the observations. It should be pointed out that for nearly all the CCMs (except for the EOF1 of CMAM), the simulated spatially standardized deviations of EOF1 and ENAD mode are smaller than those in the observation, suggesting that there exists systematic biases in these CCMs in simulating the spatial variabilities of EOF1 and ENAD mode.

Figure R4. Taylor diagram of simulated (a) EOF1 and (b) ENAD derived from CCMI1 simulations in February for period 1980-2010.

The time series of polar vortex shift and ENAD indices are also examined. Note

that, in the most simulations of 13 CCMs, there is a positive correlation between vortex shift index and ENAD index (Figure R5). This modeled positive correlation in the CCMs is in good agreement with the observations and further confirms that the ENAD mode is indeed related to the polar vortex shift. Also note that, among the models which are able to reproduce the positive correlation between the vortex shift and ENAD indices, only IPSL, EMAC-L90MA, MRI-ESM1r1, HadGEM3-ES and SOCOL3 can capture positive trends in polar vortex shift and ENAD indices. However, HadGEM3-ES and SOCOL3 could not reproduce the EOF1 spatial pattern very well (Fig. R4a).

Figure R5. Scatter plot of correlation coefficient between polar vortex shift and ENAD indices, and linear trends in 'polar vortex shift + ENAD' derived from CCM11 simulations in

February for period 1980-2010. The ‘polar vortex shift + ENAD’ is defined as the sum of normalized polar vortex shift index and normalized ENAD index to consider the overall performance on vortex shift and ENAD of CCMs. The black horizontal line represents the correlation coefficient is significant at 99% confidence, while the vertical line denotes the zero ‘polar vortex shift + ENAD’ trend.

Figure R6 shows the multi-model mean spatial patterns of EOF1 and ENAD mode, and time series of polar vortex strength, PC1, polar vortex shift as well as ENAD indices derived from CCM1-1 simulations of EMAC-L90MA, MRI-ESM1r1 and IPSL. It can be seen that the multi-model mean results indeed capture the spatial patterns of EOF1 and ENAD modes as well as positive trend of PC1 and positive trend of ENAD index in the historical period, which is consistent with the observation.

Figure R6. Spatial patterns of (a) EOF1 and (c) ENAD mode, and time series of (b) PC1 and

(d) ENAD index of multi-model mean TCO of EMAC-L90MA, MRI-ESM1r1 and IPSL in February. The blue straight lines in (b) and (d) represent linear trends in PC1 and ENAD index, respectively. The red straight lines in (b) and (d) represent linear trends in polar vortex strength and polar vortex shift indices, respectively. The correlation coefficients between PC1 and the vortex strength index, and between ENAD index and the vortex shift index are shown in the top right of (b) and (d), respectively.

Figure R7 shows the multi-model mean spatial patterns of EOF1 and ENAD modes and time series of PC1 and ENAD indices for the period 2010-2050 derived from CCMI-1 REF-C2 simulations. Note that in the future, the EOF1 show positive TCO changes over the Northern Hemisphere, with positive PC1 trend from 2010 to 2050. This positive trend may be related to the stratospheric ozone recovery in response to ODSs decline and increasing GHG gases in the future (Austin et al., 2006; Bednarz et al., 2016). In addition, the simulated ENAD pattern still shows a negative centre over the Eurasian continent and a positive one over North America. It is interesting that there exists a significant correlation between PC1 and the vortex strength, as well as a good correlation between ENAD and polar vortex shift index in the future (Fig. R7b,d), implying that the vortex shift in the future also has a contribution to the stratospheric ozone trend. The polar vortex shift index shows an increasing trend during 2010-2041 while it decreases after 2041. Given the relationship between the polar vortex shift and sea-ice changes, this phenomenon may be related to different sea-ice trends in different periods. Figure R8a indicates that there is still a negative trend in the sea-ice over the Barents and Kara seas and the polar vortex still shows a shift towards Eurasia during 2010-2041 (Fig. R7d). By contrast, the sea-ice shows an increasing trend after 2041 (Fig. R8b), presumably due to the natural variability, corresponding to a decline in polar vortex shift index during 2041-2050 (Fig. R7d). Correspondingly, the ENAD index also shows an increasing trend during 2010-2041 and decreases after 2041.

All above results further confirm the polar vortex shift could exert an important influence on the future ozone variations at NH mid- and high latitudes.

Figure R7. Spatial patterns of (a) EOF1 and (c) ENAD mode, and time series of (b) PC1 and (d) ENAD index of the averaged TCO of EMAC-L90MA, MRI-ESM1r1 and IPSL in February for the period 2010-2050. The blue straight lines in (b) and (d) represent linear trend in PC1 and ENAD during 2010-2050, respectively. The red and green straight lines in (d) denote linear trends in polar vortex shift index during 2010-2041 and 2041-2050, respectively. The correlation coefficients between PC1 and the vortex strength index, and between ENAD index and the vortex shift index are shown in the top right of (b) and (d), respectively.

Figure R8. Linear trends by percentage in multi-model mean September-October-November-December-January-February (SONDJF) SIC of EMAC-L90MA, MRI-ESM1r1 and IPSL for the period (a) 2010-2041 and (b) 2041-2050. The linear trends over the dotted regions are statistically significant at the 90% confidence level according to the Student's t-test.

References:

Austin, J., & Wilson, R. J. (2006). Ensemble simulations of the decline and recovery of stratospheric ozone. *Journal of Geophysical Research*,

Bednarz, E. M., Maycock, A. C., Abraham, N. L., Braesicke, P., Dessens, O., & Pyle, J. A. (2016). Future Arctic ozone recovery: the importance of chemistry and dynamics. *Atmospheric Chemistry and Physics*, 16(18), 12159-12176.

The CCMI results highlight the importance of ENAD in the future. However, according to the Chemistry-Climate Model Initiative (CCMI) data policy (http://www.met.reading.ac.uk/~qr903932/CCMI-website/Wordpress_PDFs/CCMI-Data-Policy_FINAL.pdf), we have to offer coauthorships to all model PIs (more than 20 co-authors) if we include the analysis of CCMI data in the revised manuscript. To send the manuscript to more than 20 PIs and ask them to read

through the paper for confirmation of their coauthorships is a big challenging task for us. In this round revision, we only keep the CCMI results in the reply files and clean up the above CCMI analysis in the manuscript. Alternatively, we show the negative correlation coefficient between ENAD and sea-ice variation and the continuing Arctic sea ice loss in near future in the main text (Supplementary Figure 7) to provide an evidence of positive ENAD trend in the future.

The introduction lacks some basic introduction to the dynamical differences between the Arctic and Antarctic polar vortices. Some of the factors alluded to arise simply from the fact that wave driving is larger in the Northern Hemisphere, making the polar vortex more perturbed than it is in the SH. The Arctic vortex is more disturbed on both intraseasonal and interannual timescales, whereas the Antarctic vortex is climatologically stronger, allowing for substantial ozone depletion. (e.g, Waugh and Polvani 2010; Solomon et al. 2014; Sheshadri et al. 2015 and probably many others, this is now almost textbook material).

Response: Thanks for your good suggestion. We have rewritten Introduction section and added information required by the reviewer in the revised paper, particularly, the first paragraph (L21-L37) is rewritten as follows:

“Stratospheric ozone protects living organisms at the Earth’s surface by strongly absorbing solar ultraviolet radiation⁵⁻⁷. It also plays an important role in modulating the global climate system by changing large-scale atmospheric circulations via its radiative impact and radiative-chemical-dynamical feedbacks, which are not only seen in the Southern Hemisphere⁸⁻¹², but also in the Northern Hemisphere¹³⁻¹⁶. Global stratospheric ozone concentrations have experienced persistent declines in response to increased ozone-depleting substances (ODSs) in the 20th century. Owing to the effects of Montreal Protocol and its amendments, the rate of stratospheric ozone depletion has recently begun to slow down¹⁷⁻²⁰. Particularly, Antarctic ozone has been reported to experience the third-stage of ozone recovery²¹. In contrast, the polar vortex in Arctic region is more dynamically disturbed by planetary waves than in the Antarctic vortex²²⁻²⁴ and complex interactions between chemical and dynamical processes at high northern latitudes

make it more challenging to identify Arctic stratospheric ozone trends²⁵⁻²⁸. Recently, some studies have argued that dynamical processes such as the Arctic polar vortex changes will play a more significant role in affecting Arctic ozone in the future; consequently, the timing of Arctic stratospheric ozone recovery is not so robust²⁸⁻³².”

All the above references have been added into the revised paper and please see the new reference list.

In looking at the polar vortex shift, did you make allowances for the fact that in about 6 years out of 10, the polar vortex underwent major warming events? Particularly when there is a “split” event, the vortex splits into two daughter vortices, the bigger one of which is over Eurasia and the smaller one of which is over North America (Matthewman et al. 2008). Perhaps what you are seeing as a trend towards a shift in the location of the vortex is simply averaging over these events? There were 12 subsequent years with SSWs starting in the early 2000’s.

Response: Zhang et al. (2016) pointed out that the shift of the polar vortex towards Eurasia is closely related to the strengthened upward wave associated with Arctic sea-ice loss in past decades. However, more upward propagating waves may not necessarily cause a sudden stratospheric warming (SSW) event. On most occasions, it may cause a weaker polar vortex, which is more likely to be displaced, rather a SSW event. From the climatological point of view, Figure R9 shows the annual occurrence of major SSWs during December, January and February for the period 1979-2009 from the statistical analysis of Tomikawa (2009). Note that SSW events in winter during the 2000s are not statistically significantly more than those in the 1980s, although there are less SSW events in the 1990s, i.e., the SSW frequency shows a decadal change with no significant trend, while the vortex area over the Eurasian continent in February shows a significant persistent positive trend since 1980 (Fig. 2d-e in the revised paper).

Figure R9. Annual time series of the SSW index. 1 and -1 represents major SSW events whether occur or not, respectively, during December, January and February for the period 1979-2009, according to the statistical results of Tomikawa (2009).

Meanwhile, Figure R10 shows time series of the duration of SSWs. Clearly there is no statistically significant trend in the SSW duration over the past decades, while the trend in the vortex area over the Eurasian continent is significant. Particularly for the period after 2000, the SSW duration shows an evident

decreasing trend, while the vortex area over Eurasia has a positive trend. These results suggest that the polar vortex shift is not necessarily related to the SSW occurrence, although some SSW events could cause the vortex shift towards Eurasia. The enhanced amplification of planetary wave associated with sea-ice loss (Zhang et al., 2016) does not necessarily imply more SSW events, i.e., there is no direct cause-and-effect relationship between SSW and long-term polar vortex shift.

Figure R10. Time series of the duration of SSWs (red line) and polar vortex shift index (see Method section). The duration of SSWs is defined as the number of days when the gradient of 10 hPa temperature between 60°–90°N in DJF becomes positive. The linear trends of two time series are also shown. Solid lines represent the corresponding linear trends which are statistically significant at the 95% confidence levels; whereas, the linear trends denoted by the dashed lines are not significant at the 90% confidence level.

As the long-term polar vortex shift towards Eurasia is not necessarily caused by major SSW events, we didn't distinguish vortex-split and vortex displacement cases in this study. In the present study, we only analysed the percentage area of well-defined polar vortex covering the Eurasian continent and its relationship with ozone changes.

Reference:

Tomikawa, Y. Persistence of Easterly Wind during Major Stratospheric Sudden Warmings. *J Climate* 23, 5258-5267 (2010).

L23 Most such studies have focused on the SH, where ozone depletion has been substantial. Ozone anomalies or trends have been quite small in the NH in comparison.

Response: We agree with your comment that the stratospheric ozone depletion in the NH is smaller than that in the SH. However, the most interesting topic is to alert the NH ozone change for the public since there are many studies reported that NH ozone depletion on human health and climate change should be closely monitored. It is estimated that UV increases resulting from a sustained 1% decrease in stratospheric ozone could cause a 2% increase in the incidence of non-melanoma skin cancer, and an approximately 0.5% increase in the incidence of cataracts (Longstreth et al., 1995). In addition, WMO (2014) reported that current total ozone column (TCO) levels at northern midlatitudes and in Arctic are on average 3.5% and 5% below the pre-1980 values, respectively, suggesting that the potentially harmful impact of NH ozone depletion on the environment is not negligible. Furthermore, recent studies have revealed the significant influence of Arctic ozone loss on the climate change in the NH. Smith and Polvani (2014), Calvo et al. (2015) and Ivy et al. (2017) showed a statistically significant Northern Hemisphere mid-high latitude surface response to low values of Arctic stratospheric ozone (ASO) with simulations and observations. Xie et al. (2016) even pointed out that the impact of ASO in the past three decades can extend to the tropics, with the ASO variations leading El Niño-Southern Oscillation (ENSO) events by about 20 months.

References:

- Calvo, N., Polvani, L. M., & Solomon, S. (2015). On the surface impact of Arctic stratospheric ozone extremes. *Environmental Research Letters*, 10(9), 1-8.
- Ivy D J, Solomon S, Calvo N and Thompson D W J 2017 Observed connections of Arctic stratospheric ozone extremes to Northern Hemisphere surface climate *Environ. Res. Lett.* 12 024004
- Longstreth, J. D., De, G., Kripke, M. L., Takizawa, Y. & van Der Leun, J. C. Effects of increased solar ultraviolet radiation on human health. *Ambio*24, 153-165 (1995).
- Smith, K. L., & Polvani, L. M. (2014). The surface impacts of Arctic stratospheric ozone anomalies. *Environmental Research Letters*, 9(7).
- van der Leun, J. C., Y., Y. Takiyawa and J. D. Longstreth (1989) *Human Health*, Chapt 2 in

Environmental effects of ozone depletion; J. C. van der Leun, M. Trevini and R. C. Worress (Eds), 11-24, United Nations Environment Programme, Nairobi, Kenya. (WMO), W. M. O. Scientific Assessment of Ozone Depletion: 2014, Global Ozone Research and Monitoring Project – Report No. 55, 416 pp., Geneva, Switzerland. (2014).
Xie, F. et al. A connection from Arctic stratospheric ozone to El Niño-Southern oscillation. Environmental Research Letters 11, 124026 (2016).

L47 EOF analysis of what? Daily ozone concentrations as a function of longitude and latitude? Was the variable of interest weighted in any way?

Response: The EOF analysis is performed on monthly TCO data as a function of longitude and latitude. The grid points in the horizontal data domain (45–90°N) were weighted by the square root of the cosine of latitude. The above information has been clarified in the Methods section.

L62 There is some longitudinal structure

Response: Thank you. We have emphasized this point as follows:

“The first leading mode accounts for 41.5% of the total variance, and shows a uniform TCO change in the mid- and high latitudes and a longitudinal structure in TCO variability.”

L117 you mean vortices

Response: Corrected.

L126 it is not active “transport” in the way that this sentence implies. The “balanced” state is that that vortex itself is located in a different position, and therefore the ozone concentrations appear shifted in location. “Transport” implies transport by the circulation.

Response: We have rephrased this expression in the revised paper as follows:

“Figure 3d–f clearly show negative dynamical ozone anomalies over the Eurasian continent, i.e., the low ozone center is also shifted along with the polar vortex shift.”

Responses to Referee 4

This study can be considered as the “part 2” of a recent paper published by the same group of authors in *Nature Clim. Change* (Zhang et al. 2016). Therefore, its background mainly relies on this previous “part 1” investigation. In short, sea ice depletion over the North Pole coupled with an increased snow precipitation over the north Asia worked together in shifting the polar vortex over the Eurasian continent and away from North America during the last decades. With this in mind, the new study focuses on the ozone changes induced by this shift. By analyzing ozone reanalysis and simulations carried out with SLIMCAT model the authors attribute these changes to transport of ozone-poor air from Arctic and to local chemical processes caused by the transport of active chlorine species towards lower latitudes.

Overall, the study is well planned, results are convincing and it provides a useful contribution to evaluate the ozone temporal evolution/recovery at northern high latitudes. Nevertheless, in the present form this work can hardly be published in a high ranking journal while I find it more suitable for JGR or ACP. Personally, I would find more convenient having included some of the present findings within the part 1, but I understand the need of splitting the work, so I don't blame the authors. However, having already achieved the main results in the part 1, the additional findings presented in this new paper look somewhat ordinary and far from being very intriguing. The authors state that “The question remains whether this shift of the polar vortex induced by climate change can change trace gas distributions, especially the ozone distribution in the Northern Hemisphere during late winter”. In the light of the part 1 of the study, it seems rather plausible that ozone would follow the vortex shift so one can expect what the authors described here. Obviously, the study has some merit e.g. the assessment of the ozone transport with respect to the local chemical processes, but I think that at the present stage this is not enough to warrant publication on *Nature Communication* and some further work should be carried out. Also, the conclusions of this study don't look novel (e.g. “The results suggest that the future ozone recovery in late-winter would be sensitive not only to ODSs change but also to the polar vortex change”) and I believe that a broader paper instead of a short communication could better address the issues here mentioned.

I actually don't have specific suggestions on how making this work more

appealing, nevertheless I still see that the analysis is basically limited to SLIMCAT and MSR2 datasets. Due to the limited temporal extension of the time series and the expected large internal climate variability in the northern high latitudes, can an additional analysis based on climate models (either from the CCMI or CMIP5) be useful in this context? Moreover, since things seem to be somewhat clear for the “historical” period, I would find even more remarkable the investigation of the future ozone response under different RCP scenarios.

Response: We thank the reviewer for his/her comments. In the original version of this study, we mainly analyzed the historical influence of polar vortex shift and gave a perspective that the ENAD could delay the ozone recovery in the future. We agree with the reviewer that it would be unique and important to include additional analysis based on climate models for this work. That is, we used chemistry-climate models (CCMs) to predict the polar vortex shift and ENAD changes in the future.

First, we checked the performance of the models in Chemistry-Climate Model Initiative Phase-1 (CCMI-1) in simulating the two leading modes of Northern Hemisphere TCO and polar vortex shift in historical period for the purpose of picking out suitable CCMs to simulate future ENAD changes. To evaluate the performance of 13 CCMs in simulating historical spatial patterns of EOF1 and ENAD, Taylor diagram (Figure R11) is plotted, which shows how closely a spatial pattern from models matches that from observation. With regards to EOF1 of the TCO (Fig. R11a), the spatial patterns in five CCM simulations (i.e., IPSL, EMAC-L90MA, MRI-ESM1r1, NIWA-UKCA and CNRM-CM5-3) are the closest to the observations (REF in Fig. R11a). For the ENAD mode (Fig. R13b), the spatial patterns in the five simulations still keep high correlations with the observations. It should be pointed out that for nearly all the CCMs (except for the EOF1 of CMAM), the simulated spatially standardized deviations of EOF1 and ENAD mode are smaller than those in the observation, suggesting that there exists systematic biases in these CCMs in simulating the spatial variabilities of EOF1 and ENAD mode.

Figure R11. Taylor diagram of simulated (a) EOF1 and (b) ENAD derived from CCM11 simulations in February for period 1980-2010.

The time series of polar vortex shift and ENAD indices are also examined. Note that, in the most simulations of 13 CCMs, there is a positive correlation between vortex shift index and ENAD index (Figure R12). This modeled positive correlation in the CCMs is in good agreement with the observations and further confirms that the ENAD mode is indeed related to the polar vortex shift. Also note that, among the models which are able to reproduce the positive correlation between the vortex shift and ENAD indices, only IPSL, EMAC-L90MA, MRI-ESM1r1, HadGEM3-ES and SOCOL3 can capture positive trends in polar vortex shift and ENAD indices. However, HadGEM3-ES and SOCOL3 could not reproduce the EOF1 spatial pattern very well (Fig. R11a).

Figure R12. Scatter plot of correlation coefficient between polar vortex shift and ENAD indices, and linear trends in ‘polar vortex shift + ENAD’ derived from CCM1 simulations in February for period 1980-2010. The ‘polar vortex shift + ENAD’ is defined as the sum of normalized polar vortex shift index and normalized ENAD index to consider the overall performance on vortex shift and ENAD of CCMs. The black horizontal line represents the correlation coefficient is significant at 99% confidence, while the vertical line denotes the zero ‘polar vortex shift + ENAD’ trend.

Figure R13 shows the multi-model mean spatial patterns of EOF1 and ENAD mode, and time series of polar vortex strength, PC1, polar vortex shift as well as ENAD indices derived from CCM1-1 simulations of EMAC-L90MA, MRI-ESM1r1 and IPSL. It can be seen that the multi-model mean results indeed capture the spatial patterns of EOF1 and ENAD modes as well as positive trend of PC1 and positive trend of ENAD index in the historical period, which is

consistent with the observation.

Figure R13. Spatial patterns of (a) EOF1 and (c) ENAD mode, and time series of (b) PC1 and (d) ENAD index of multi-model mean TCO of EMAC-L90MA, MRI-ESM1r1 and IPSL in February. The blue straight lines in (b) and (d) represent linear trends in PC1 and ENAD index, respectively. The red straight lines in (b) and (d) represent linear trends in polar vortex strength and polar vortex shift indices, respectively. The correlation coefficients between PC1 and the vortex strength index, and between ENAD index and the vortex shift index are shown in the top right of (b) and (d), respectively.

Figure R14 shows the multi-model mean spatial patterns of EOF1 and ENAD modes and time series of PC1 and ENAD indices for the period 2010-2050 derived from Chemistry-Climate Model Initiative Phase-2 (CCMI-2) simulations. Note that in the future, the EOF1 show positive TCO changes over the Northern Hemisphere, with positive PC1 trend from 2010 to 2050. This positive trend may

be related to the stratospheric ozone recovery in response to ODSs decline and increasing GHG gases in the future (Austin et al., 2006; Bednarz et al., 2016). In addition, the simulated ENAD pattern also shows a negative centre over the Eurasian continent and a positive one over North America. It is interesting that there exists a significant correlation between PC1 and the vortex strength, as well as a good correlation between ENAD and polar vortex shift index in the future (Fig. R14 b,d), implying that the vortex shift in the future also has a contribution to the stratospheric ozone trend. The polar vortex shift index shows an increasing trend during 2010-2041 while it decreases after 2041. Given the relationship between the polar vortex shift and sea-ice changes, this phenomenon may be related to different sea-ice trends in different periods. Figure R15a indicates that there is still a negative trend in the sea-ice over the Barents and Kara seas and the polar vortex still shows a shift towards Eurasia during 2010-2041 (Fig. R14d). By contrast, the sea-ice shows an increasing trend after 2041 (Fig. R15b), presumably due to the natural variability, corresponding to a decline in polar vortex shift index during 2041-2050 (Fig. R14d). Correspondingly, the ENAD index also shows an increasing trend during 2010-2041 and decreases after 2041.

All above results further confirm the polar vortex shift could exert an important influence on the future ozone variations at NH mid- and high latitudes.

Figure R14. Spatial patterns of (a) EOF1 and (c) ENAD mode, and time series of (b) PC1 and (d) ENAD index of the averaged TCO of EMAC-L90MA, MRI-ESM1r1 and IPSL in February for the period 2010-2050. The blue straight lines in (b) and (d) represent linear trend in PC1 and ENAD during 2010-2050, respectively. The red and green straight lines in (d) denote linear trends in polar vortex shift index during 2010-2041 and 2041-2050, respectively. The correlation coefficients between PC1 and the vortex strength index, and between ENAD index and the vortex shift index are shown in the top right of (b) and (d), respectively.

Figure R15. Linear trends by percentage in multi-model mean September-October-November-December-January-February (SONDJF) SIC of EMAC-L90MA, MRI-ESM1r1 and IPSL for the period (a) 2010-2041 and (b) 2041-2050. The linear trends over the dotted regions are statistically significant at the 90% confidence level according to the Student's t-test.

References:

Austin, J., & Wilson, R. J. (2006). Ensemble simulations of the decline and recovery of stratospheric ozone. *Journal of Geophysical Research*,

Bednarz, E. M., Maycock, A. C., Abraham, N. L., Braesicke, P., Dessens, O., & Pyle, J. A. (2016). Future Arctic ozone recovery: the importance of chemistry and dynamics. *Atmospheric Chemistry and Physics*, 16(18), 12159-12176.

The CCMI results highlight the importance of ENAD in the future. However, according to the Chemistry-Climate Model Initiative (CCMI) data policy (http://www.met.reading.ac.uk/~qr903932/CCMI-website/Wordpress_PDFs/CCMI-Data-Policy_FINAL.pdf), we have to offer coauthorships to all model PIs (more than 20 co-authors) if we include the analysis of CCMI data in the revised manuscript. To send the manuscript to more than 20 PIs and ask them to read

through the paper for confirmation of their coauthorships is a big challenging task for us. In this round revision, we only keep the CCMI results in the reply files and clean up the above CCMI analysis in the manuscript. Alternatively, we show the negative correlation coefficient between ENAD and sea-ice variation and the continuing Arctic sea ice loss in near future in the main text (Supplementary Figure 7) to provide an evidence of positive ENAD trend in the future.

Then, is MSR2 the best observational reference for this study? You should justify this choice. Recently, many combined O3 datasets spanning more than three decades are become available. Although they mostly rely on solar occultation instruments and therefore are characterized by a sparse and infrequent sampling at high latitudes, the new SBUV v8.6 datasets could still be useful. Moreover, since both the data assimilation of MSR2 as well as the SLIMCAT model are driven by ERA-interim, it could be interesting including an additional independent assimilated ozone dataset. A good candidate can be the Bodeker's NIWA ozone dataset. Finally, as ERA-interim assimilates also ozone fields, perhaps, despite its limitations, ERA-interim ozone could even be useful for checking the consistency of these results.

Response: Thanks, this is a very useful point. We have checked the consistency of the results based on the comments. Figures R16 a-b show spatial patterns of 'ENAD mode' (EOF2+EOF3) derived from NIWA data and ERA-Interim reanalysis data, respectively. The ENAD modes in these two data sets both show a negative centre of TCO variation over the Eurasian continent and a positive centre over the North American continent, similar with those in MSR2 data and SLIMCAT simulation in the original manuscript. The spatial patterns of ENAD modes also resemble the regression patterns of NIWA and ERA-Interim TCO against the vortex shift index, respectively (Figs. R16 c-e). Furthermore, both the ENAD indices (PC2+PC3) derived from NIWA and ERA-Interim data (Figs. R16 g-h) have statistically significant correlation coefficients between vortex shift index. The above results indicate that the ENAD mode of TCO associated with the polar vortex shift also exists in NIWA and ERA-Interim data.

We extended the analysis period of SLIMCAT simulation to 1980-2015 (Fig. R16 right panel). The spatial pattern of ENAD model in the longer SLIMCAT

simulation is also similar to the regression of TCO on the vortex index (Figs. R16 c,i) and time series of ENAD mode is closely related to vortex shift index (Fig. R16 f). These results confirm that the ENAD mode of TCO caused by the polar vortex shift is also evident during the longer time period from 1980 to 2015.

Figure R17a shows that the ENAD-like mode can be also found in the clear-sky ultraviolet (UV) radiation over the Northern Hemisphere and there is a significant relationship between PC2+PC3 of UV index and the polar vortex shift index (Fig. R17b). The ENAD-like mode in UV index also resembles the regression pattern of UV index against the vortex shift index (Fig. R17c), suggesting that the polar vortex shift can exert a significant influence on the clear-sky UV radiation over the Northern Hemisphere. The above analysis has been added in the revised paper.

Figure R16. (a–c) Spatial patterns of ‘ENAD mode’ (EOF2+EOF3) and (d–f) time series of normalized ENAD index (PC2+PC3) (blue lines) of February TCO over 45–90°N derived from (left) NIWA data during 1980-2012, (middle) ERA-Interim data during 1980-2015 and (right) SLIMCAT simulations during 1980-2015. Time series of vortex shift index (see Methods section, red lines) are overplotted in (d–f). The percentage of explained variance is shown in the top right of (a–c), and the correlation coefficients between PC2+PC3 and the vortex shift index are shown in the top right of (d–f). Linear trends of TCO regressed on the vortex shift index are shown derived from (g) NIWA data, (h) ERA-Interim data and (i) SLIMCAT simulation output. The linear trends over the dotted regions are statistically significant at the 90% confidence level according to the Student’s t-test. The minimum latitude of polar stereographic projections is 45°N. The values in February 1987, 2006, 2009 and 2013 are not calculated because the polar vortex broke up and its shape was distorted in these months (see Methods section).

Figure R17. As in Fig. R16, but for the surface clear-sky erythemal UV (280-400 nm) index

derived from MSR2 data.

Reviewers' comments:

Reviewer #1 (Remarks to the Author):

The authors have addressed all my comments. I support publication.

A side comment: the further methods you introduced for assessing the vortex edge are also frame dependent, as Potential Vorticity. Regarding frame dependence, the authors may want to look into the following reference: Truesdell C, Noll W. 2004. The Non-Linear Field Theories of Mechanics.

Reviewer #2 (Remarks to the Author):

The authors have implemented all my points and from what I see they have dealt, comprehensively and elegantly, with all the other reviewers' comments as well. In my opinion the manuscript is ready for publication in NCOMMS.

I would also like to offer my view on the points raised during the review (by other reviewers) and the revision (by the authors) regarding this manuscript's work/content in relation to previous or future publications:

The manuscript in its revised form is a complete and valuable piece of work, highlighting the past-present link of the Arctic ozone hole with the polar vortex shift and it can stand alone as a publication with its current content. Specifically:

- a) It is an important follow-up on the previous publication (Nature Clim. Change, Zhang et al. 2016) which only focused on the dynamics.
- b) The additional work of the comparison with the CCMI results and projections should be elaborated and become a follow-up publication for the future N.H. ozone changes and links with circulation.

Reviewer #3 (Remarks to the Author):

I continue to think that the results presented in this manuscript are not particularly novel, and that this paper was written more from the desire to publish another manuscript than from any real need to split this work. (Particularly if this is aimed at specialists in the field -- specialists in the field would expect that ozone concentrations co-vary with the location of the vortex.)

Some comments:

1. You could calculate EOFs of anything and call them modes (which is not even accurate terminology, since EOFs are statistical patterns and not modes of an underlying dynamical system).
2. I'm not sure what you mean by evidence that this "mode" could delay future ozone recovery, to me it just seems like you are saying that future ozone recovery will be set by both future emissions and the dynamics of the polar vortex (both of which are pretty obvious statements). I have now read this manuscript twice, and it hasn't convinced me that ENAD has implications for understanding anything really. To repeat, it is not surprising at all that there is a correlation between your ENAD and the shift of the vortex edge in the CCMI models either (one expects ozone concentrations to co-vary with the vortex location -- the opposite conclusion would be surprising).

3. Are daily data not available?

4. Does Figure R10 show the duration of SSWs in February or during all months? (This paper and Zhang et al. 2016 only discuss February so the other months are not relevant. If you are convinced that what you are seeing has nothing to do with SSWs, or, indeed, final warmings in February, please clarify why you think so in the manuscript. Many "events" in the stratosphere push the vortex off the pole and towards Eurasia for a couple of weeks at a time.)

L49 more sensitive than when (and why?)?

L52-55 only in February

Reviewer #4 (Remarks to the Author):

The revised version of the work has been greatly improved and I recognize that the authors made the previously missing step toward a very good work. All my previous comments have been satisfactorily addressed by analyzing additional observations (e.g. NIWA, ERA-interim) and simulations (CCMs). In particular, it is relevant that the authors' findings have been confirmed by a large ensemble of CCMs for the "historical" period and that the same models suggest that the polar vortex shift will influence also future ozone variations. Further interesting additional outcomes are:

- 1) the clear relationship between the polar vortex shift and sea-ice changes in the future simulations
- 2) the fact that also an independently assimilated ozone dataset (i.e. NIWA) as well as ERA-interim ozone present the same features highlighted in MLR2
- 3) the surface UV changes at high latitudes coupled to ozone variations.

Overall, the paper is now suitable for publication although I still note that not including the new (important) results related to the CCMs analysis lowers the quality of the work. I see the issue related to the data policy. However, since this further analysis has been provided under the revision phase in order to address the comments of this reviewer, I believe that you can propose to the PIs only an explicit acknowledgment or even an authorship subjected to a quick confirmation (1 week?) due to editorial needs.

In the 2nd revised paper, we added the PIs of all the Chemistry-Climate Model Initiative (CCMI) models as co-authors, according to the editor's and reviewers' comments and suggestions. The authors have reviewed the manuscript and confirmed their authorships. Our detailed replies to the 3rd reviewer's comments are given below.

Responses to Referee 3

I continue to think that the results presented in this manuscript are not particularly novel, and that this paper was written more from the desire to publish another manuscript than from any real need to split this work. (Particularly if this is aimed at specialists in the field -- specialists in the field would expect that ozone concentrations co-vary with the location of the vortex.)

Some comments:

1. You could calculate EOFs of anything and call them modes (which is not even accurate terminology, since EOFs are statistical patterns and not modes of an underlying dynamical system).

Response: Although the EOF analysis is a mathematical tool, it is commonly used to detect leading spatial-temporal patterns in meteorological fields. The ENAD pattern derived from EOF analysis highly resembles the TCO pattern projecting on polar vortex shift index, suggesting that the ENAD mode indeed has a physical meaning.

Furthermore, the reliability of the ENAD mode by EOF analysis is further confirmed in this revision. We found this mode is persistent and stationary in either ozone depletion period or ozone recovery period. We removed the linear trend of TCO time series before EOF analysis and divided TCO time series into two periods, i.e., 1980-2000 and 2000-2012. Figure R1 shows the ENAD pattern derived from SLIMCAT TCO data for the periods 1980-2000 and 2000-2012 with the EESC trend removed in the SLIMCAT simulations. In both periods 1980-2000 and 2000-2012, the principal component of ENAD has a significant correlation coefficient (0.8) with the polar vortex shift index, indicating that the ENAD pattern is closely related to the polar vortex shift in both periods. Note that the ENAD pattern during 1980-2000 explains 18.4% of total TCO variance, while the ENAD during 2000-2012 can explain 60.3% of TCO variance, suggesting that the ENAD plays a more important role in Arctic TCO variation after 2000. Furthermore, both the positive center and the negative center of the anomalous TCO pattern for the period 2000-2012 are shifting more northward compared with those during 1980-2000. This result implies that the EOF analysis of TCO without EESC trend in the two periods can also capture the TCO changes associated with the polar vortex shift. The analysis result from the MSR2 TCO data with EESC trend removed is similar with the SLIMCAT result (not shown).

All above results indicate that the EOF patterns of the TCO over the Northern

Hemisphere have a physical basis and that the ENAD pattern is a good indicator of the TCO variation attributed to the polar vortex shift.

Figure R1. ENAD spatial pattern and time series of SLICMAT TCO with EESC trend removed for the period 1980-2000 and 2000-2012. The blue and red lines in (c) and (d) represent ENAD index and polar vortex shift index, respectively.

2. I'm not sure what you mean by evidence that this "mode" could delay future ozone recovery, to me it just seems like you are saying that future ozone recovery will be set

by both future emissions and the dynamics of the polar vortex (both of which are pretty obvious statements). I have now read this manuscript twice, and it hasn't convinced me that ENAD has implications for understanding anything really. To repeat, it is not surprising at all that there is a correlation between your ENAD and the shift of the vortex edge in the CCM1 models either (one expects ozone concentrations to co-vary with the vortex location -- the opposite conclusion would be surprising).

Response: Thanks for your comments. Previous studies documented that climate change influences Arctic stratospheric ozone mainly through modulating the polar vortex strength and Arctic temperature. Our present study proposed a new viewpoint that the sea-ice loss associated with Arctic climate change could cause the polar vortex shift and further slow regional total ozone recovery over the Eurasian continent. The ENAD mode can help us highlight the above phenomenon more clearly and an increasing ENAD trend suggests the Arctic sea-ice loss may continuously slow the TCO recovery in the near future, which has an implication for understanding long-term stratospheric ozone changes over the northern middle and high latitude.

3. Are daily data not available?

Response: Figure R2a shows daily mean polar vortex strength in 2001 and PC1 of TCO 5 days later. There is a high correlation (0.97) between them, suggesting that the changes in polar vortex strength could influence the EOF1 5 days later. Furthermore, the correlation coefficient between daily vortex shift index and

PC2+PC3 (i.e., ENAD) 5 days later is 0.63 (significant at 99% confidence level, Fig. R2b), implying the relationship between ENAD and the polar vortex shift also exists in daily data. Similar significant correlations between PC1 and vortex strength, ENAD and vortex shift index have also been found in daily data during the two other polar vortex shift cases (i.e., 2008 (c-d) and 2010 (e-f)).

Figure R2. (left) PC1 (red lines) of daily TCO derived from SLIMCAT simulation and polar vortex strength index 5 days ago (blue lines) in 2001, 2008 and 2010. (right) PC2+PC3 (i.e. ENAD) of daily TCO (red lines) and polar vortex shift index 5 days ago (blue lines) in 2001, 2008 and 2010. The TCO data are

derived from SLIMCAT simulation.

4. Does Figure R10 show the duration of SSWs in February or during all months?

(This paper and Zhang et al. 2016 only discuss February so the other months are not relevant. If you are convinced that what you are seeing has nothing to do with SSWs, or, indeed, final warmings in February, please clarify why you think so in the manuscript. Many “events” in the stratosphere push the vortex off the pole and towards Eurasia for a couple of weeks at a time.)

Response: Figure R3 shows the SSW duration and polar vortex shift in February.

Note that there is no significant correlation between them. Furthermore, the

SSW duration shows no significant trend while the polar vortex shift in February

has an increasing trend. These results suggest that SSW may be not the direct

cause for the polar vortex shift and its associated ENAD pattern.

Figure R3. Duration of the sudden stratospheric warming (SSW) events (blue) and polar vortex shift index (red) in February.

L49 more sensitive than when (and why?)?

Response: We have rephrased this sentence in the revised paper as follows:

“Recent studies have speculated that the Northern Hemisphere stratospheric ozone evolutions in the future may be more likely to be affected by the changes in strength of Arctic polar vortex than by chemical forcings²⁸⁻³⁰ as the implementation of Montreal Protocol undermines the contribution of ODS to ozone trend.”

L52-55 only in February

Response: Thank you for your comments. We have emphasized that the polar vortex shift occurs in late winter (February).